



# Linking extreme rainfall to suspended sediment fluxes in a deglaciating Alpine catchment

Amalie Skålevåg[1,*], Lena Katharina Schmidt[1], Nele Eggers[1,2,3], Jana Tjeda Brettin[1], Oliver Korup[1,4], and Axel Bronstert[1]

[1]Institute of Environmental Science and Geography, University of Potsdam, Potsdam, Germany
[2]Institute of Physics and Astronomy, University of Potsdam, Potsdam, Germany
[3]Alfred-Wegener-Institute for Polar and Marine Research, Potsdam, Germany
[4]Institute of Geosciences, University of Potsdam, Potsdam, Germany
[*]Now at: Norwegian Meteorological Institute, Oslo, Norway

**Correspondence:** Amalie Skålevåg (amalie.skalevag@met.no)

**Abstract.** Sediment transport in high-Alpine environments is undergoing a fundamental shift as glaciers retreat and extreme precipitation events become more frequent. Understanding how these changes influence suspended sediment yields (SSY) is critical for predicting future sediment dynamics, water quality, and geomorphic evolution in mountain catchments. This study investigates the role of extreme precipitation in driving suspended sediment export in the rapidly deglaciating, nested Alpine

5   catchments of Tumpen-Ötztal and Vent-Rofental in Austria. We examine how precipitation and rainfall intensity, frequency, and duration influence suspended sediment yields and concentrations. Using a 21-year dataset of high-resolution precipitation and a multi-scale detection approach, we identify extreme precipitation events and analyse their characteristics and contribution to sediment transport. Events are classified based on their temporal characteristics, distinguishing between sub-daily and long-duration extremes, and spatial scale, distinguishing between catchment-wide and grid-scale extremes. We also evaluate

10   the influence of precipitation uncertainties. Our findings show a significant increase in the frequency of extreme precipitation events and their contribution to annual SSY. Sub-daily extremes, primarily convective summer storms, generate disproportionately high sediment fluxes due to their localized and intense rainfall. Sediment transport during long-duration extremes responds more strongly to increases in event rainfall intensity and totals. Despite an increasing trend in extreme-precipitation-driven sediment fluxes, annual SSY remains stable in Tumpen-Ötztal but declines in Vent-Rofental, suggesting that extreme-

15   precipitation-driven transport may partially offset, but not fully replace, glacier-driven sediment supply. As climate projections indicate a continued rise in extreme precipitation, particularly at sub-daily scales, Alpine catchments may develop increasingly flashier sediment regimes in the future. However, long-term reductions in glacier-driven sediment supply will likely lead to declining annual sediment yields. These findings highlight the need for continued monitoring and study of changing precipitation dynamics, sediment transport, and paraglacial landscape evolution in high-Alpine environments.



## 1 Introduction

Heavy precipitation is projected to increase in both frequency and intensity with rising global temperatures (Madsen et al., 2014; Vergara-Temprado et al., 2021; Fowler et al., 2021). In high mountain areas like the European Alps, where precipitation patterns are strongly influenced by topography, changes in precipitation are spatially heterogeneous and differ between seasons (Giorgi et al., 2016; Menegoz et al., 2020; Brönnimann et al., 2018). At the same time, the ongoing degradation of the mountain cryosphere, in particular glacier mass loss, alters sediment availability and export over decadal scales (Schmidt et al., 2022, 2023; Zhang et al., 2022b; Delaney and Adhikari, 2020). In combination, these changes to precipitation patterns and the mountain cryosphere have affected hydrological and sediment transport regimes of rivers (Zhang et al., 2023; Kormann et al., 2016) and measurably increased the amount of fluvial sediment exported from some high-mountain areas (Li et al., 2021a; Vergara et al., 2022; Costa et al., 2018; Zhang et al., 2022a; Delaney and Adhikari, 2020; Vergara et al., 2024) . Elevated sediment loads in rivers can negatively impact downstream communities, infrastructure, and ecosystems, particularly by altering flood frequencies, degrading water quality, impairing hydro-power production, and disrupting aquatic habitats (Adler et al., 2022; Huss et al., 2017; Scheurer et al., 2009; Li et al., 2022).

Peak fluvial sediment fluxes in mountainous regions are often associated with extreme precipitation (Skålevåg et al., 2024; Li et al., 2021b; Rainato et al., 2021; Wulf et al., 2012; Himmelstoss et al., 2024; Scorpio et al., 2022). Rainstorms may cause runoff and erosion; slope wash from rainsplash, sheet flow, rill erosion, or gullying; and trigger mass movements such as debris flows and landslides, thus mobilizing sediment that eventually enters the channel network (Wischmeier and Smith, 1978; Beylich et al., 2017; Scorpio et al., 2022; Hirschberg et al., 2019; Leonarduzzi et al., 2017; Himmelstoss et al., 2024; Rom et al., 2023). Streamflow peaks in response to rainfall also enhance channel erosion via bed incision and bank erosion (Rainato et al., 2021; Scorpio et al., 2022). Another control on sediment dynamics in Alpine catchments during rainstorms is the increase in functional sediment connectivity (Scorpio et al., 2022; Buter et al., 2022; Himmelstoss et al., 2024), which elevates sediment fluxes by better coupling hillslopes to the channel network. Given the projected increases in summer convective rainfall at high elevations in the European Alps (Giorgi et al., 2016; Dallan et al., 2024) and the waning influence of glaciers on annual sediment transport (Schmidt et al., 2023, 2024), the timing and frequency of extreme precipitation are likely to be of increased relevance for fluvial sediment transport in cryospheric basins.

Yet, at least two confounding factors complicate the quantitative assessment of extreme-precipitation-driven sediment transport in high mountain areas. First, the scarcity of weather stations and the complex topography of mountainous terrain means that estimates of precipitation at high elevations is associated with high uncertainties, which add to the intrinsically high errors tied to rare events by virtue of extreme-value theory. Second, both glacial processes and deglaciation remain an important, but rarely systematically captured, control on sediment production and transport (Schmidt et al., 2022; Li et al., 2024). Paraglacial landscapes might respond differently to future increases in extreme precipitation than unglaciated basins, because reaches downstream of glaciers host higher amounts of unconsolidated sediments and sparse vegetation cover. Projections suggest that, with the decreasing influence of glaciers (Schmidt et al., 2024) precipitation-driven sediment fluxes could become more dominant, sediment-transport regimes become more flashier and more dependent on erosive rainfall events (Zhang et al., 2022b).





Hence, by studying the influence of extreme precipitation in the current transient state and analysing whether sediment export associated with extreme precipitation is already changing, we may glean important insights about the hydro-geomorphic future of Alpine rivers.

In this study, we employ a multi-scale detection approach based on extreme-value statistics to assemble a catalogue of extreme precipitation events in a catchment in the Ötztal Alps, Austria. By using an hourly, 1-km gridded precipitation product

for catchment-averaged and grid-scale maximum precipitation time series, we identify extreme precipitation peaks at multiple temporal and spatial scales. Each detected event is quantified in terms of precipitation intensity, duration, seasonality, spatio-temporal pattern, and mass of suspended sediment exported from the catchment. We wish to understand the response of suspended sediment yield to precipitation extremes, by addressing the following objectives:

- to quantify fluvial sediment responses to precipitation extremes, including differences between types of precipitation
extremes; and

- to identify trends in precipitation- and extreme-precipitation-driven contributions to annual fluvial sediment yield.

## 2 Study area and data

Our study area is the valley of Ötztal in Tyrol, Austria, which is located in the comparatively dry region of the Central Alps relative to the rest of the European Alps (Fig. 1a). The valley has been the focus of several hydrometeorological and glaciological

studies and has unique long-term observations (see Strasser et al., 2018). About 10% of the catchment is currently glaciated (Schmidt et al., 2022; Buckel and Otto, 2018) but the glaciated area is rapidly decreasing and the glacier ice volume is projected to be reduced to 4-20% by 2100 (Hanzer et al., 2018). The hydrological regime is dominated by snow and ice melt (Strasser et al., 2018; Kormann et al., 2016). The highest amounts of both precipitation and rainfall occur in the summer months between June and August (Fig. 2). Outside of the main melt season (May-October) little to no rainfall occurs. The timing of peak

seasonal flow differs between individual catchments, with a later peak at higher elevations, i.e. in the Vent-Rofental catchment (Fig. 2; Schmidt et al., 2022). The suspended sediment flux is also highest during the summer months and the seasonal cycle is fairly synchronous between the main catchment and the high-elevation sub-catchment Vent-Rofental (Fig. 2; Schmidt et al., 2022).

### 2.1 Streamflow and suspended sediment

The monitoring of riverine sediments in Ötztal is part of Austria's national strategy to assess changes in sediment dynamics due to factors like deglaciation and land-use change (for details, see Lalk et al., 2014). The two gauging stations used in this study, Tumpen and Vent, are operated by the Hydrographic Service of Tyrol (HD-Tirol). Tumpen station (931 m asl, 46.85797°N, 10.91049°E) is situated on the Ötztaler Ache, a few kilometers upstream of the outlet of Ötztal valley. The Tumpen-Ötztal catchment covers 785.5 km$^2$ which is most of Ötztal valley and spans almost 3000 m of elevation from 931 to 3772 m asl



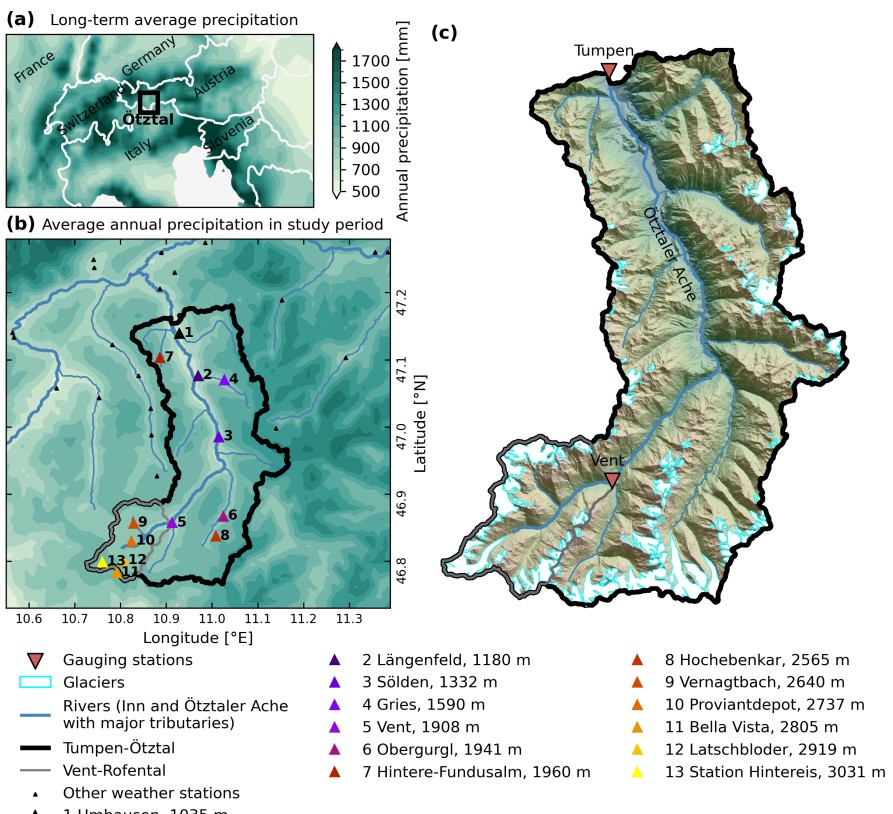

**Figure 1.** Average annual precipitation of the European Alps (a) and the study area, Ötztal (b). Precipitation data (1801-2014) from HISTALP (GeoSphere Austria, 2020) show the drier central Alps, where the Ötztal is located (a). Annual precipitation in Tumpen-Ötztal during the study period (2004-2024) from INCA (GeoSphere Austria, 2024a) ranges between 632 mm at the valley floor to 1223 mm along the eastern catchment boundary (b). Most of the 13 weather stations in the study area are located along the main valley (e.g. Umhausen, Längenfeld and Sölden) or in the Vent-Rofental catchment (e.g. Proviantdepot). Station elevation is given in m asl. The topography of Ötztal is steep with the exception of river plains in the lower half of the central valley (c). Glaciers are concentrated in the Stubai Alps along the catchments eastern border and the Ötztal Alps to the south (c).

(Schmidt et al., 2022). Vent station (1891 m asl, 46.85691°N, 10.91093°E) is located in the village of Vent and drains the Rofental valley. The 98 km$^2$ Vent-Rofental catchment is in the headwaters of Tumpen-Ötztal.

Suspended sediment concentrations (SSC) have been monitored since 2006 at both stations using optical infrared turbidity sensors calibrated with manual samples. At Tumpen, turbidity is continuously monitored throughout the year, while at Vent measurements are paused in winter (November-April) to protect the equipment from damage by ice. Sediment transport is

considered negligible during this period (Fig. 2).



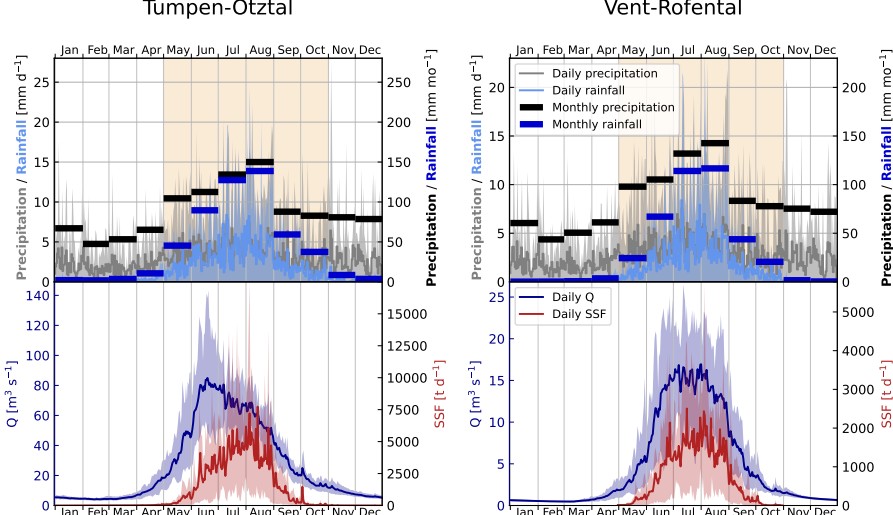

**Figure 2.** Seasonal cycle of precipitation, rainfall, streamflow (Q), and suspended sediment flux (SSF). Top panels show daily (grey lines) and monthly (black horizontal bars) precipitation totals based on data from 2004 to 2024, while daily (light blue lines) and monthly (blue horizontal bars) rainfall totals are averages of 2011-2024 (data described in Sec. 2.2.1) with 10-90% percentile ranges (shaded areas). Bottom panels show 2006-2022 averages (lines) and 10-90% percentile ranges (shaded areas) of daily average Q and total daily SSF (data described in Sec. 2.1). Horizontal bars show average monthly precipitation and rainfall totals. Precipitation follows the same seasonal cycle in both catchments. Little to no rainfall occurs outside of the melt season (May-October; highlighted in yellow). The wettest months both in terms of precipitation and rainfall are the summer months (June-August).

For this study we used 15-min time series of streamflow, $Q_t$, and SSC, $\mathrm{SSC}_t$. Time series of suspended sediment flux $\mathrm{SSF}_t$ in t $15\mathrm{min}^{-1}$ is calculated by multiplying $Q_t$ and $\mathrm{SSC}_t$ in each time step. These data are available between 2006 and 2022, meaning that data on fluvial sediment transport is only available for 16 of the 21 years of the study period (2004-2024).

## 2.2 Precipitation data

The main precipitation data used in this study are hourly precipitation grids from GeoSphere Austria (GSA) which are supplemented by daily and hourly precipitation from weather stations in and around Ötztal (Fig. 1b).

### 2.2.1 INCA

To estimate catchment-wide precipitation and rainfall we used gridded precipitation and temperature data from the analysis product of GSA's Integrated Nowcasting through Comprehensive Analysis (INCA) system (Haiden et al., 2011; GeoSphere
Austria, 2024a). This blended product integrates observations, weather radar, numerical weather prediction (NWP) outputs, and topographical information into hourly 1-km grids for all of Austria.



Precipitation estimates are based on a radar composite from four to five C-band radars supplemented with data from neighbouring countries, and calibrated with rain-gauge measurements from approximately 250 weather stations and elevation effects (Haiden et al., 2011). Observed precipitation is interpolated onto the INCA grid with inverse distance weighting (IDW). To-

pographical errors in the radar composite is corrected by applying a climatological scaling and spatially rescaling it using the latest observations. Finally, the twice adjusted radar field is combined with the interpolated observations. For temperature estimation, INCA employs a three-dimensional analysis method, in which NWP outputs are adjusted with measured temperatures (see Kann et al., 2009; Haiden et al., 2011). The accuracy of INCA estimates can vary, particularly in complex terrain, with an average error of 50-100% in the 15-minute precipitation grids and 1.0 to 1.5 °C in the temperature grids (Haiden et al., 2011).

Hourly INCA precipitation grids from 15 March 2011 to 12 December 2024 (GeoSphere Austria, 2024a) were merged with hourly grids aggregated from 15-min resolution grids from 1 January 2004 to 14 March 2011 to create a unified hourly precipitation dataset from 2004 and 2024. We performed a simple quality check on the unified dataset, removing negative values and checking each time step with grid-scale precipitation $> 100 \ \mathrm{mm \ h^{-1}}$. In the latter case, we removed seven time steps where these high precipitation rates were clearly data artefacts.

Hourly rainfall grids were estimated by calculating the precipitation phase with the routine from the snow-hydrological model openAMUNDSEN (Hanzer et al., 2024; Strasser et al., 2024) using INCA temperature grids. The resulting liquid precipitation fraction grids were multiplied with INCA precipitation to obtain hourly rainfall grids. As rainfall estimates rely on temperature grids (GeoSphere Austria, 2024a), which begin on 15 March 2011, hourly rainfall grids are only available for the same time period as temperature (i.e. March 2011 to December 2024).

Hourly time series of catchment-averaged precipitation, $P_t$, and grid-scale maximum precipitation, $I_t$, for the Ötztal-Tumpen catchment were calculated from INCA precipitation grids by taking the mean and maximum of all grid cells within the catchment for each time step. Similarly, we calculated hourly catchment-averaged rainfall, $\mathrm{RF}_t$, by averaging INCA rainfall grids over the Ötztal-Tumpen catchment for each time step.

### 2.2.2 Station data

We collected daily and sub-daily precipitation measurements from multiple weather stations in and around Ötztal (Fig. 1, Tab. A1). Most stations are operated by GSA or the Hydrographic Service of Tyrol (HD-Tirol) and tend to be located at lower elevations on the valley floor. Therefore we supplemented with high-elevation stations from the Department of Geography (UIBK-GEOG) and the Department of Atmospheric and Cryospheric Sciences (ACINN) at the University of Innsbruck, as well as the Vernagtbach station operated by the Bavarian Academy of Sciences (BADW) (see Strasser et al., 2018; Warscher

et al., 2024). The weather stations have varying coverage during the study period (Appendix, Fig. A1, see Tab. A1 for a complete list of weather stations).

Most of the stations have already undergone initial quality checks by the providers in terms of the precipitation data, except for the ACINN stations, which feature raw data. For these stations we performed visual quality checks of all data to remove implausible values. For comparison with the gridded INCA data, we aggregated the measurements to hourly and daily resolution

where applicable with the criteria that the aggregation interval must contain at least 90% valid data.





## 3  Methods

### 3.1  Uncertainty analysis of INCA

Gridded precipitation products in mountainous regions have limited accuracy (e.g. Prein and Gobiet, 2017; Zandler et al., 2019; Deng et al., 2024; Sleziak et al., 2023) due to the strong influence of topography on precipitation, an observation bias towards

lower elevations, and challenging conditions for radar (e.g. beam shielding) (Germann et al., 2006). Even in the mountainous parts of the INCA domain (i.e. Austrian Alps), the high density of weather stations is somewhat biased towards elevations below 2000 m asl (Haiden et al., 2011). Taking advantage of the higher density of rain gauges in Ötztal also at high elevations (Strasser et al., 2018; Warscher et al., 2024), we can estimate the uncertainty of hourly and daily INCA precipitation with our assembled rain gauge data in and around Ötztal (Tab. A1) using four metrics.

The mean error (ME, Eq. 1) and the root-mean squared error (RMSE, Eq. 2) are calculated for each station by comparing the observed precipitation of a station $x_i^{\mathrm{obs}}$ with the predicted INCA precipitation at the grid cell in which the station is located $x_i^{\mathrm{INCA}}$

$$\mathrm{ME} = \frac{1}{N} \sum_{i=1}^{N} (x_i^{\mathrm{INCA}} - x_i^{\mathrm{obs}}) \tag{1}$$

$$\mathrm{RMSE} = \sqrt{\frac{1}{N} \sum_{i=1}^{N} (x_i^{\mathrm{INCA}} - x_i^{\mathrm{obs}})^2} \tag{2}$$

where $N$ is the total number of time steps indexed by $i$ with both valid INCA and observation values. The ME indicates whether INCA tends to over- or under-estimate precipitation at the station, whereas the RMSE quantifies the overall error magnitude of INCA (Wilks, 2019).

To estimate the ability of INCA to capture the occurrence of precipitation we classified each time step of $x_i^{\mathrm{obs}}$ and $x_i^{\mathrm{INCA}}$

into a "precipitation" and "no-precipitation" category, using a threshold of 0.1 mm h$^{-1}$ for the hourly data, and 1 mm d$^{-1}$ for the daily data. The latter is the definition of a "wet day" commonly used in climate indices (Zhang et al., 2011). Next, we computed contingency tables (see Gold et al., 2019) listing the number of hits (true positives) $a$, misses (false negatives) $b$, false positives $c$, and true negatives $d$. From these four possible outcomes we estimated (1) the accuracy (Acc, Eq. 3) and (2) the frequency bias (FB, Eq. 4) (Wilks, 2019; Gold et al., 2019).

$$\mathrm{Acc} = \frac{a+d}{N} \tag{3}$$

$$\mathrm{FB} = \frac{a+c}{a+b} \tag{4}$$

The Acc is the fraction time steps in which INCA correctly predicts the occurrence of precipitation. The FB quantifies whether INCA tends to over- or under-estimate the occurrence of precipitation.



### 3.1.1 Annual-based uncertainty analysis

To quantify how uncertainties in INCA precipitation estimates may have changed during the study period, we conducted an annual-based analysis. We calculated RMSE of daily precipitation for each year and station separately. Next, we calculated annual RMSE averages from all available stations within Tumpen-Ötztal. Of particular interest is whether precipitation and heavy precipitation days ($>10\,\mathrm{mm\,d^{-1}}$) are over- or under-predicted, as this may affect the detection of precipitation extremes. Hence we also calculated the FB of these two quantities for each year of the study period.

### 3.2 Multi-scale detection of extreme precipitation events

To assemble a catalogue of extreme precipitation events, we device a multi-scale detection approach based on extreme value statistics that allows for detection of precipitation extremes at multiple temporal and spatial scales. We use catchment-averaged, $P_t$, and grid-scale maximum precipitation, $I_t$, to represent the spatial scales of the catchment- and grid-scale precipitation (Fig. 3). Extremes in $P_t$ will tend to represent catchment-wide heavy precipitation such as frontal precipitation. Extremes in $I_t$ will track grid-scale extremes, generally convective cells. Using a duration-dependent generalised extreme value (d-GEV) distribution, we estimate intensity-duration-frequency (IDF) curves of $P_t$ and $I_t$ during May-October for Tumpen-Ötztal. With the IDF curves we set detection thresholds for each duration, extract peaks, isolate the associated precipitation event, and merge any duplicated or overlapping events. The procedure is described in detail below.

For each of the two time series, $P_t$ and $I_t$, we fit a d-GEV distribution (Koutsoyiannis et al., 1998) which allows us to calculate IDF curves from a single extreme value distribution (see Ulrich et al., 2020; Fauer et al., 2021) reducing the total number of parameters required (Ulrich et al., 2020). We use the the R-package `IDF` (Fauer et al., 2017) with the options allowing multi-scaling and curvature for small durations (see Fauer et al., 2021). The d-GEV distributions were fitted to annual May-October precipitation block maxima, $\mathbf{M} = (m_{dj})$, using the maximum likelihood method. We calculated $\mathbf{M}$ for each of the precipitation time series $P_t$ and $I_t$. For each duration $d$, $\mathbf{M}$ were calculated by applying a $d$-moving average to the precipitation time series and extracting the maximum value $m$ during May-October of each year $j$. We considered the durations 1, 2, 3, 6, 12, 24, 48, and 72 h. We restricted $\mathbf{M}$ to May-October, when most sediment is exported and rainfall is highest (Fig. 2). Outside of this season precipitation predominantly falls and accumulates as snow and is therefore not relevant for our study.

Detection thresholds, $u$, for each $d$ and spatial scale (i.e. $I_t$ or $P_t$) were set at the 0.8 exceedence probability quantile of their respective IDF curves, which corresponds to a return period of 1.25 years. With this choice of $u$ we should capture the major precipitation events each year for the spatial and temporal scales under consideration and thus ensure a sufficiently large catalogue of precipitation extremes.

Next, we detected and isolated extreme peaks in the precipitation time series. From $d$-moving-averaged $P_t$ and $I_t$ we extracted peaks above $u$ and their timing, $t_{\mathrm{peak}}$ (Fig. 3). We detected event peaks for all durations in $P_t$ but only for durations $\leq$ 24 h in $I_t$, under the assumption that grid-scale extremes will generally be convective events that last less than 24 h. For each extreme peak, we isolated the associated precipitation event by searching forward and backward in time from $t_{\mathrm{peak}}$ to identify




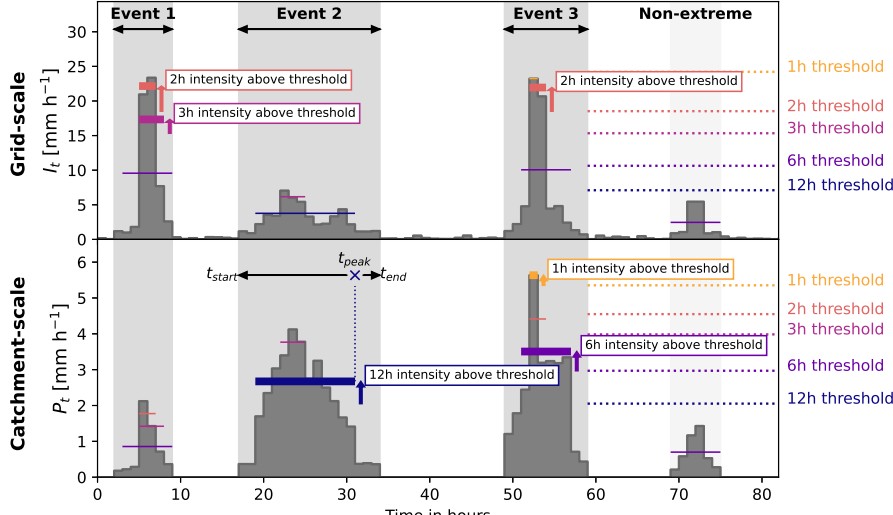

**Figure 3.** Illustration of multi-scale detection of precipitation extremes (synthetic time series). Extreme precipitation peaks above the detection thresholds are detected both on the grid-scale with the grid-scale maximum precipitation time series $I_t$ (top) and on the catchment-scale with the catchment-averaged precipitation time series $P_t$ (bottom). Peaks exceeding the detection thresholds (only sub-daily durations show in this figure) are identified (thick coloured lines with labels and arrows showing threshold exceedance), while those below are ignored (thin coloured lines). From the timing of the detected peak $t_{\mathrm{peak}}$, the detection algorithm searches forward and backward in time to identify the start $t_{\mathrm{start}}$ and end $t_{\mathrm{end}}$ of the event (grey area). Event 1 shows a case where extreme peaks of two durations were detected from $I_t$ (grid-scale) but the event was not extreme at the catchment scale. Event 2 has a single extreme peak at the catchment scale. Event 3 has extreme peaks both at the grid- and catchment-scale. The non-extreme towards the end of the time series shows a case where precipitation did not exceed the thresholds at any duration and spatial scale.

when it started and stopped raining. The event start time, $t_{\mathrm{start}}$, was defined as the first time step before $t_{\mathrm{peak}}$ that satisfied the criteria

$$200 \qquad \sum_{t=t_{\mathrm{start}}-1}^{t_{\mathrm{start}}+1} P_t < 0.1 \ \mathrm{mm \ h^{-1}} \tag{5}$$

or

$$\sum_{t=t_{\mathrm{start}}-1}^{t_{\mathrm{start}}+1} I_t < 1 \ \mathrm{mm \ h^{-1}} \tag{6}$$

depending on in which precipitation time series the extreme peak was detected. The event end time, $t_{\mathrm{end}}$, was defined as the first time step after $t_{\mathrm{peak}}$ that satisfied the criteria in Eq. 5 and 6 ($t_{\mathrm{start}}$ substituted with $t_{\mathrm{end}}$).

205     Many precipitation extremes were detected at multiple scales, i.e. these events exceeded $u$ for several durations, or for both $P_t$ and $I_t$. These events are either duplicates (i.e. same $t_{\mathrm{start}}$ and $t_{\mathrm{end}}$) or temporally overlapping that we merged iteratively in the following order:



1. duplicated or overlapping events with peaks for the same duration, detected in both $P_t$ and $I_t$;

2. events with identical $t_{\mathrm{end}}$;

3. events with identical $t_{\mathrm{start}}$; and

4. any remaining overlapping events.

Overlapping events were merged by updating $t_{\mathrm{start}}$ and $t_{\mathrm{end}}$ to the earliest $t_{\mathrm{start}}$ and the latest $t_{\mathrm{end}}$, so that the merged event encompassed the timespan of all events being merged. Each precipitation event was tagged with the durations and spatial scales at which it was extreme. All detected events were manually checked by visually evaluating their accumulated precipitation maps and comparing their time series with station observations. Events with implausible values or precipitation patterns were checked thoroughly and removed if it was judged that the event was a data artefact or mistaken detection.

### 3.3 Event precipitation characteristics

For each precipitation extreme, we calculate a set of characteristics based on Leonarduzzi et al. (2017), which quantify rainfall and precipitation amounts and intensity (Tab. 1). In addition, we calculated the average precipitation area $A_{\mathrm{precip}}$ in order to estimate the catchment area affected by a precipitation extreme. We calculated $A_{\mathrm{precip}}$ by determining the fraction of catchment area receiving $> 0.1\,\mathrm{mm}$ precipitation for each time step, before averaging over the entire event duration.

We further categorised each event according to the spatial and temporal scale of the set of extreme peaks detected. An event detected from $P_t$ or both $P_t$ and $I_t$ was categorised as a *catchment-scale* event, while one detected only from $I_t$ was categorised as a *grid-scale* event. *Sub-daily* extremes are events with peaks detected above 1 to 12 h thresholds. *Long-duration* extremes are events with at least one extreme peak detected above a 24, 48, or 72 h threshold. Hence, the temporal scale does not refer to the overall duration of the precipitation event, i.e. $D$ in Tab. 1, but the set of extreme peaks detected (Fig. 3).

#### 3.3.1 Uncertainty analysis

To gauge the uncertainty in the precipitation intensity and amount of each detected event, we calculated the RMSE of $P_{\mathrm{tot}}$ and $I_{\mathrm{max}}$ for each event. This analysis was based on the 10 stations within Ötztal with hourly precipitation measurements (see Tab. A1). Due to the varying temporal extents of the station data, we first calculated the event-based RMSE for each station then averaged over all stations to obtain one RMSE value for $P_{\mathrm{tot}}$ and $I_{\mathrm{max}}$ respectively. This ensures that each station is weighted equally which prevents biasing the estimate towards the lower elevation stations that have more observations.

### 3.4 Fluvial sediment response to extreme precipitation events

To estimate the fluvial sediment response to precipitation events we first delineated hydrological events in $Q_t$ using local-minima hydrograph separation (Sloto and Crouse, 1996). Following Tsyplenkov et al. (2020) we use a 21-hour search window, which is suitable for Ötztal's glacial hydrological regime (Skålevåg et al., 2024).





**Table 1.** Metrics quantifying the characteristics of the extreme precipitation events

| Metric | Description | Equation | Unit |
|---|---|---|---|
| $P_\mathrm{tot}$ | Total cumulative catchment-averaged precipitation | $P_\mathrm{tot} = \sum_{t=t_\mathrm{start}}^{t_\mathrm{end}} P_t$ | mm |
| $\mathrm{RF}_\mathrm{tot}$ | Total cumulative catchment-averaged rainfall | $\mathrm{RF}_\mathrm{tot} = \sum_{t=t_\mathrm{start}}^{t_\mathrm{end}} \mathrm{RF}_t$ | mm |
| $D$ | Event duration | $D = t_\mathrm{end} - t_\mathrm{start}$ | h |
| $f_\mathrm{liquid}$ | Fraction of liquid precipitation | $f_\mathrm{liquid} = \frac{\mathrm{RF}_\mathrm{tot}}{P_\mathrm{tot}}$ | - |
| $I_\mathrm{max}$ | Maximum intensity of grid-scale maximum precipitation | $I_\mathrm{max} = \max\{I_t : t_\mathrm{start} \leq t \leq t_\mathrm{end}\}$ | mm h$^{-1}$ |
| $P_\mathrm{max}$ | Maximum intensity of catchment-averaged precipitation | $P_\mathrm{max} = \max\{P_t : t_\mathrm{start} \leq t \leq t_\mathrm{end}\}$ | mm h$^{-1}$ |
| $\mathrm{RF}_\mathrm{max}$ | Maximum intensity of catchment-averaged rainfall | $\mathrm{RF}_\mathrm{max} = \max\{\mathrm{RF}_t : t_\mathrm{start} \leq t \leq t_\mathrm{end}\}$ | mm h$^{-1}$ |
| $P_\mathrm{mean}$ | Mean intensity of catchment-averaged precipitation | $P_\mathrm{mean} = \frac{P_\mathrm{tot}}{D}$ | mm h$^{-1}$ |
| $\mathrm{RF}_\mathrm{mean}$ | Mean intensity of catchment-averaged rainfall | $\mathrm{RF}_\mathrm{mean} = \frac{\mathrm{RF}_\mathrm{tot}}{D}$ | mm h$^{-1}$ |

The resulting hydrological event catalogue was compared to the detected extreme precipitation events, matching hydrological events to each precipitation extreme: All hydrological events that overlap with a precipitation event, i.e. begin or end between $t_\mathrm{start}$ and $t_\mathrm{end}$, are assigned to that event. We discarded all cases, in which the first matched hydrological event began >3 h

before $t_\mathrm{start}$, or where the last matched hydrological event began <1 h before the end of the precipitation event.

The fluvial sediment response window of an extreme precipitation event was defined as the time window from $t_\mathrm{start}$ to the end of the last matched hydrological event $t_\mathrm{hydro,end}$. For each precipitation extreme we calculated the mass of suspended sediment exported, i.e. suspended sediment yield SSY (t):

$$\mathrm{SSY} = \sum_{t=t_\mathrm{start}}^{t_\mathrm{hydro,end}} \mathrm{SSF}_t \tag{7}$$

To take into account the varying event durations we also calculate the average suspended sediment flux:

$$\mathrm{SSF}_\mathrm{mean} = \mathrm{mean}\{\mathrm{SSF}_t : t_\mathrm{start} \leq t \leq t_\mathrm{hydro,end}\} \tag{8}$$





We also extracted the peak SSC concentration during the sediment response window of each precipitation extreme:

$$\mathrm{SSC_{max}} = \max\{\mathrm{SSC}_t : t_{\mathrm{start}} \leq t \leq t_{\mathrm{hydro,end}}\} \tag{9}$$

### 3.5 Contribution of precipitation-driven events to annual sediment yield

To quantify the contribution of precipitation-driven sediment transport to annual SSY, we conducted an inverse analysis in which we classified all hydrological events based on the associated precipitation. We categorised hydrological events with influence of *extreme precipitation* if they matched with precipitation extremes in Sec. 3.4. Of the remaining hydrological events, those with an average precipitation intensity of $>0.1$ mm h$^{-1}$ were categorised as *non-extreme precipitation* and the rest as *no precipitation*.

For each hydrological event we calculated SSY, SSF, $P_{\mathrm{tot}}$, and $\mathrm{RF}_{\mathrm{tot}}$, substituting $t_{\mathrm{start}}$ and $t_{\mathrm{end}}$ with the start and end times of the hydrological event (see Eq. 7-9 and in Tab. 1). Next, we calculated the contribution to annual SSY of each hydrological event under the influence of extreme, non-extreme, and no precipitation. Due to the high inter-annual variability of annual SSY we also calculated the fraction of annual SSY exported during each precipitation influence event class. Using a Mann-Kendall (MK) test (Kendall, 1970; Mann, 1945) with a 5 % significance level we detected significant annual trends, and estimated their
magnitude with Thiel-Sen slope (Sen, 1968; Theil, 1950).

### 3.6 Fraction of precipitation-driven suspended sediment spikes

Given the strong influence of melt-driven sediment transport in the study area, not all sediment discharge events with high SSC are linked to (extreme) precipitation. In another inverse analysis we extracted hydrological events with high peak SSC and classified them as influenced by extreme, non-extreme, no precipitation as in Sec. 3.5. We defined these *suspended sediment*
*spikes* as hydrological events with $\mathrm{SSC_{max}}$ above the 90th percentile of $\mathrm{SSC}_t$, $P_{90}(\mathrm{SSC}_t)$ after Skålevåg et al. (2024). For both catchments we counted the number of such events affected by either extreme or non-extreme precipitation.

## 4 Results

### 4.1 INCA uncertainty

The uncertainty analysis of daily and hourly INCA precipitation shows that INCA tends to overestimate the precipitation
amount with an average ME of 0.2 mm d$^{-1}$ and 0.01 mm h$^{-1}$ for daily and hourly precipitation respectively. The average RMSE of hourly precipitation is 0.5 mm h$^{-1}$ and is unrelated to altitude (Fig. A3d, Appendix). The RMSE of daily precipitation is 1.0 to 2.5 mm d$^{-1}$ at stations below 1750 m asl compared to 1.0 to 6.2 mm d$^{-1}$ at higher elevations (Fig. A3c, Appendix). At both, the hourly and daily scale, the INCA data are highly accurate (Acc$>0.9$) concerning precipitation up to about 2500 m asl (Fig. A3e-f, Appendix). At higher elevations, this accuracy drops to between 0.8 and 0.9 for precipitation
hours and between 0.66 and 0.93 for precipitation days. Overall, the frequency of precipitation hours and days is overestimated by INCA with FB generally above 1 (Fig. A3g-h, Appendix). However, at three stations, Proviantdepot, Latschbloder, and



Station Hintereis, located at high elevations (2737 to 3031 m als) in the inner Rofental, INCA under-predicts the occurrence of precipitation at both the daily and hourly scale.

## 4.2 Extreme event catalogue and characteristics

A total of 169 extreme precipitation events were identified with the multi-scale detection approach. Three events were removed during the visual check, as the precipitation maps revealed that only one or two grid cells received very high amounts of precipitation while neighbouring cells did not receive any and no precipitation was recorded at the weather stations; these events were likely data artefacts.

  Of the 166 events compiled in the final catalogue, 62 were detected from $I_t$, 78 from $P_t$, and 26 from both time series.
Only 41 events (25%) were detected with a single threshold, while the majority were all either extreme at multiple temporal or spatial scales. The events ranged from intense short duration extremes to long-duration events with high precipitation totals (Fig. 4b). The average event duration was 24 h with the shortest event lasting 5 h and the longest 128 h (5.3 days). The highest $I_{\max}$ was 88 mm h$^{-1}$ with an average of 19 mm h$^{-1}$ (2019-g, Fig. 4b). The largest event in terms of precipitation amount recorded a $P_{\mathrm{tot}}$ of 106 mm (2023-g, Fig. 4b) compared to the highest RF$_{\mathrm{tot}}$ of 97 mm. On average the total precipitation and
rainfall amount was 24 and 16 mm respectively.

  The 75 extremes detected with sub-daily thresholds ($\leq$ 12 hours) tended to have more intense precipitation (Fig. 4f) consisting mostly of rainfall (95 % on average). They occurred mainly between June and August, which is when precipitation and rainfall amounts are highest (Fig.2). Two examples of sub-daily extremes (Fig. 4a;d) highlight how localised precipitation was, with much the catchment receiving little to no precipitation. The 91 long-duration extremes had higher precipitation totals
(Fig. 4g) that affected larger areas (Fig. 4e), occurred throughout May-October without any particular seasonality, and with higher fractions of snowfall (45 % on average).

  The 104 catchment-scale events had low precipitation intensities (Fig. 4b,f) but affected a larger catchment area (Fig. 4e) with high precipitation totals (Fig. 4g). In contrast, the 62 grid-scale precipitation extremes affected a smaller catchment area (Fig. 4e), had high $I_{\max}$ (Fig. 4f), and precipitation totals of mostly <20 mm (Fig. 4g).

The annual frequency of precipitation extremes varied over the study period with events occurring more frequently towards the end (Fig. 5a), displaying a significant increasing trend (MK-test, 5% significance level). For the first few years of the study period there were 3-5 events per year and in these years May-October precipitation was lower than the latter part of the study period. From 2010 onwards followed a few years with high inter-annual variability in event numbers, varying between 4 and 16 events per year. In the final part of the study period, from 2015 onwards, the annual occurrence of precipitation extremes was
at a higher level with a minimum of 8 events per year. In general, there were more events in years with higher May-October precipitation, with annual event counts and May-October precipitation totals being significantly correlated, $r = 0.65$.

  Average annual RMSE of daily precipitation was higher in the first half of the study period (median: 2.4 mm d$^{-1}$) and dropped to a lower level (median: 1.4 mm d$^{-1}$) between 2016 and 2024 (Fig. 5a). In both periods daily precipitation is overestimated, but with a higher median annual mean error of 0.24 mm in 2004-2016 compared to 0.14 mm from 2017





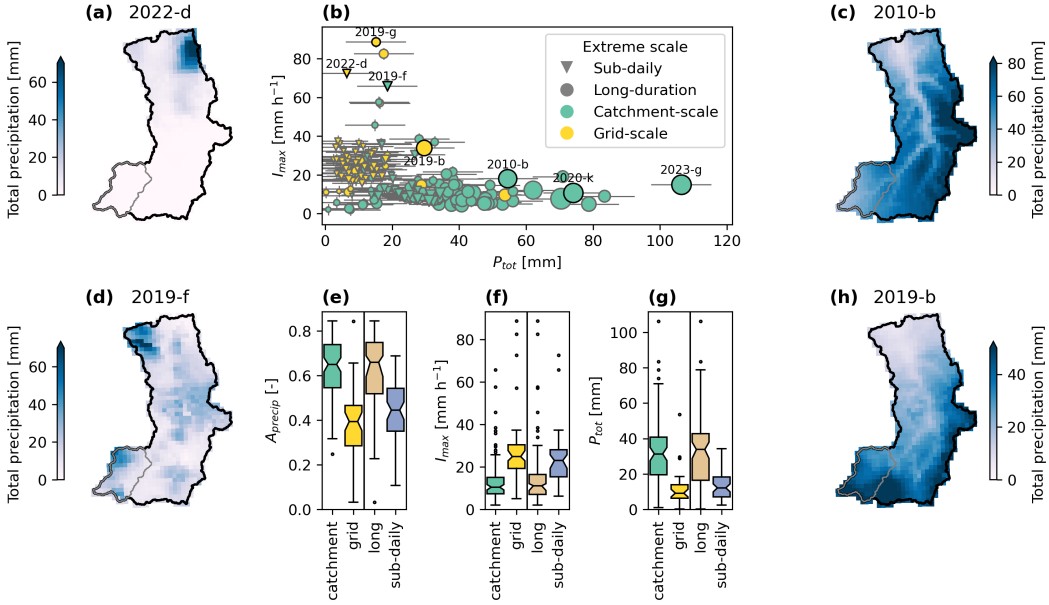

**Figure 4.** Overview of precipitation extremes detected with the multi-scale approach (b). Events range from grid-scale events of high intensity to catchment-scale events with high precipitation totals. Dot size indicates the duration of events. Error bars show the average RMSE of $P_{\text{tot}}$ and $I_{\text{max}}$ across all events and stations. Total accumulated precipitation maps of selected events (highlighted in b) show examples of a sub-daily, grid-scale event (a), sub-daily, catchment-scale event (d), long-duration, catchment-scale event (c), and long-duration grid-scale event (h), in addition to the event with the highest $P_{\text{tot}}$ (2023-g) and $I_{\text{max}}$ (2019-g). Event 2020-k exported a high amount of suspended sediment. Boxplots of average precipitation area (e), peak 1-hour catchment maximum precipitation intensity $I_{\text{max}}$ (f), and total cumulative precipitation $P_{\text{tot}}$ (g) highlight differences between catchment- and grid-scale events (left) and events with long and short extreme durations (right). Boxplot notches are 95% bootstrap confidence intervals for the median based on 1000 randomisations. The whiskers extend from the box to the most distant data point within 1.5 times the inter-quartile range (IQR) from the box.

onwards. Heavy precipitation days with more than $10 \text{ mm d}^{-1}$ had a tendency to be overpredicted, except in 2004 and 2015 where they were under-predicted. The years with the highest frequency bias are 2007, 2012, 2013, and 2014.

About two thirds of precipitation extremes occurred during the summer months July and August with the lowest occurrences in May, September and October (Fig. 5b). Events with a high liquid fraction, i.e. mainly rainfall, were concentrated in the months June, July and August. Mid-season precipitation extremes tend to have higher intensities and shorter durations, while

events with longer duration and lower intensity occurred evenly throughout the year. Events with higher amounts of snowfall (liquid fraction generally below 0.5) mainly occurred in the colder months of May, June, September and October.

### 4.3 Precipitation characteristics and sediment response

Of all precipitation event characteristics, the three rainfall characteristics show the strongest and consistently positive significant correlation with each of the sediment response variables (Tab. 2). The two characteristics describing precipitation intensity,





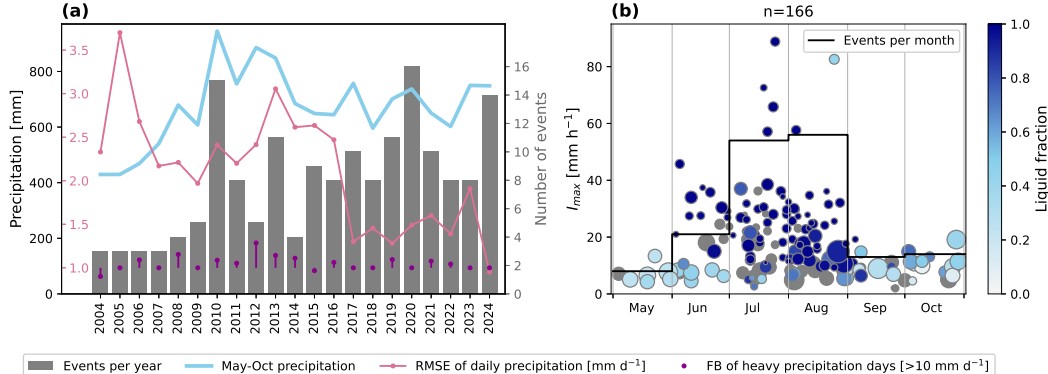

**Figure 5.** Annual (a) and seasonal (b) occurrence of extreme precipitation events. Number of events per year and May-October precipitation (a) are lower in the first six years of the study period compared to the later years. The annual median RMSE of daily precipitation across all stations within the study area show higher uncertainties in INCA for the first 13 years (a). The frequency bias (FB) of heavy precipitation days in INCA is also higher in the first decade (a). Most precipitation extremes occur during July and August (b) when events consist mostly of liquid precipitation. Dot sizes are proportional to the total event precipitation. Grey dots are events with unknown liquid fraction (2004-2010).

**Table 2.** Linear correlation coefficients between precipitation event characteristics and $\log_{10}$-transformed sediment response variables. Statistical significant correlations (5% significance level) have an asterisk (*).

|  | $\mathrm{SSY}^{\mathrm{a}}$ | $\mathrm{SSF}_{\mathrm{mean}}{}^{\mathrm{a}}$ | $\mathrm{SSC}_{\mathrm{max}}{}^{\mathrm{a}}$ |
|---|---|---|---|
| $P_{\mathrm{tot}}$ | -0.08 | -0.32 * | -0.15 |
| $P_{\mathrm{mean}}$ | 0.18 * | 0.09 | 0.22 * |
| $P_{\mathrm{max}}$ | 0.35 * | 0.26 * | 0.39 * |
| $I_{\mathrm{max}}$ | 0.30 * | 0.43 * | 0.37 * |
| $\mathrm{RF}_{\mathrm{tot}}$ | 0.51 * | 0.27 * | 0.39 * |
| $\mathrm{RF}_{\mathrm{max}}$ | 0.63 * | 0.58 * | 0.67 * |
| $\mathrm{RF}_{\mathrm{mean}}$ | 0.69 * | 0.66 * | 0.68 * |
| $D$ | -0.19 * | -0.42 * | -0.26 * |
| $A_{\mathrm{precip}}$ | -0.21 * | -0.41 * | -0.24 * |

[a] $\log_{10}$-transformed

$I_{\mathrm{max}}$ and $P_{\mathrm{max}}$, show significant but somewhat weaker positive correlation. Event duration and average precipitation area are negatively correlated with sediment response magnitude.

The relationship between event rainfall and suspended sediment shows clear differences between sub-daily and long-duration extreme precipitation events (Fig. 6). The sub-daily extremes have an overall weaker positive relationship with rainfall intensity and amount. The difference is particularly pronounced for $\mathrm{RF}_{\mathrm{max}}$ and SSY and $\mathrm{SSF}_{\mathrm{mean}}$ (Fig. 6b,f). Here, SSY and $\mathrm{SSF}_{\mathrm{mean}}$





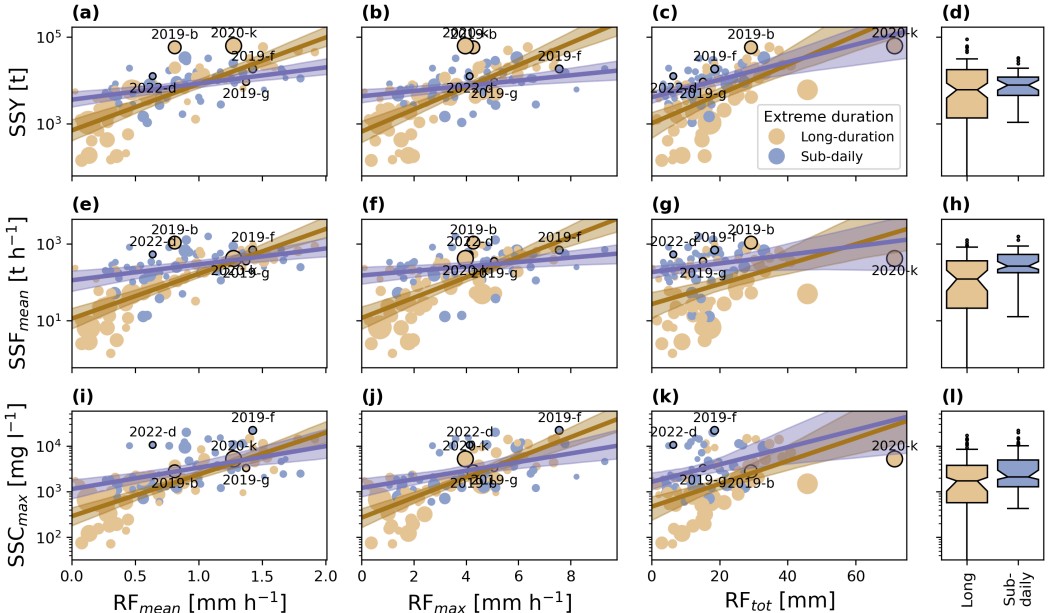

**Figure 6.** Suspended sediment response to precipitation extremes (109 events between 2011-2022) in terms of their mean intensity of catchment-averaged rainfall $RF_{mean}$(a,e,i), maximum intensity of catchment-averaged rainfall $RF_{max}$ (b,f,j), and total cumulative catchment-averaged rainfall $RF_{tot}$ (c,g,k). Dot size indicates the duration of events. Lines are means from linear regression, and shaded areas are 95% bootstrap confidence intervals based on 1000 randomisations. Labelled events are the same as in Fig. 4. See Fig. 4 for details on boxplot configuration.

markedly increase with rainfall intensity for long-duration extremes, whereas for sub-daily extremes, increases in rainfall intensity barely have an effect. Sub-daily extremes overall have higher $SSF_{mean}$ compared to long-duration extremes (Fig. 6h) and somewhat although not significantly higher $SSC_{max}$ (confidence intervals of medians overlap, Fig. 6l). In terms of total exported suspended sediment mass there is little difference between the two categories of precipitation extremes, except a larger spread for long-duration extremes (Fig. 6d).

## 4.4 Contribution of precipitation to annual sediment yield

Between 2006 and 2022, both the total mass and the fraction of suspended sediment exported during precipitation extremes increased significantly in both catchments (Fig. 7a-d). The fraction of annual SSY exported during extreme precipitation-driven hydrological events increased at about 1 % per year in both catchments (Fig. 7a,c). There are no significant annual trends in suspended sediment exported during hydrological events with non-extreme or no precipitation. Moreover, in both catchments the sediment mass exported outside of precipitation extremes (Fig. 7b,d) follows a similar trend to the total annual SSY (Fig. 7a,c). In Tumpen-Ötztal, the sediment mass exported outside of extreme precipitation events was largely unchanged



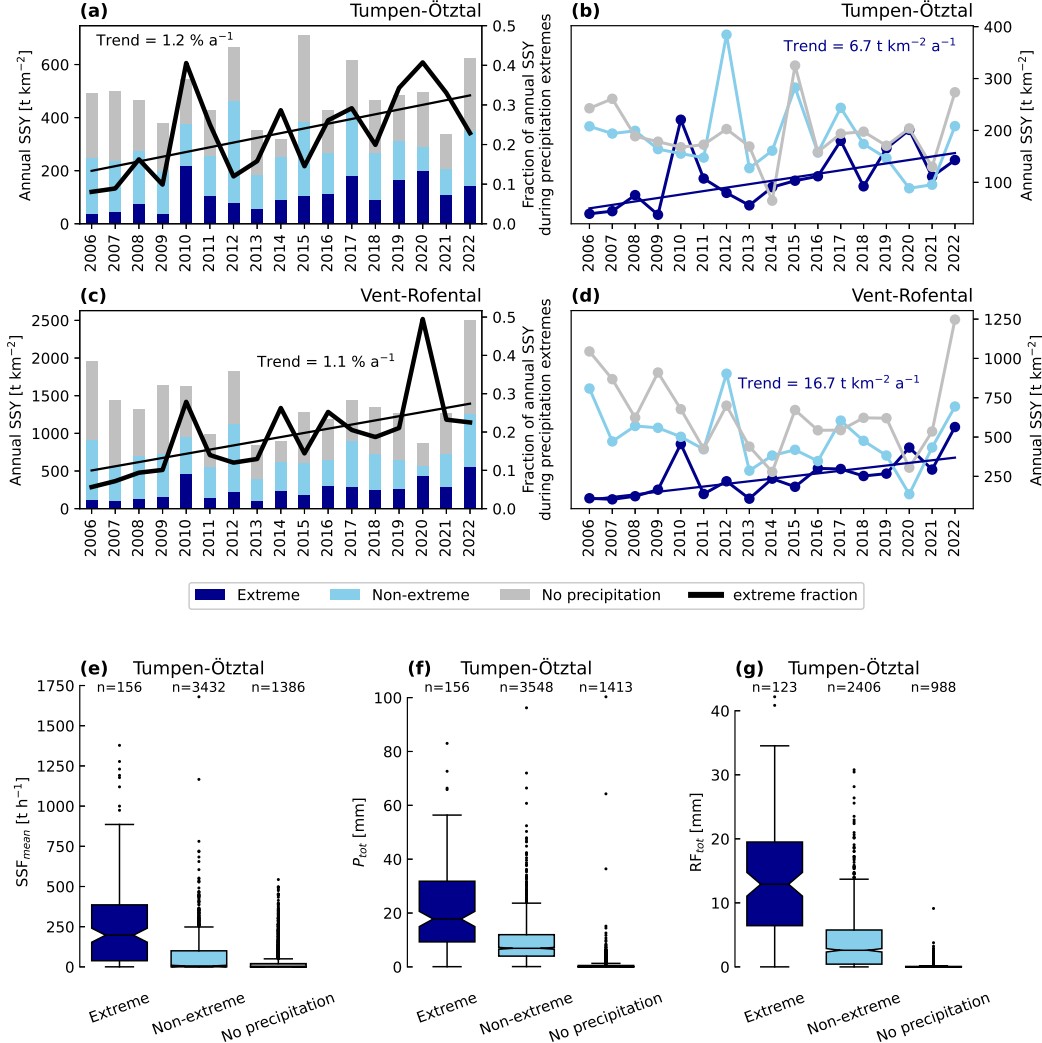

**Figure 7.** Annual suspended sediment yield (SSY) categorised by precipitation type during a hydrological event with the contribution of each type between 2006 and 2022 for Tumpen (a-b) and Vent (c-d). The fraction (thick black lines) and total amount of sediment (colour-coded bars) exported during precipitation extremes significantly increases between 2006 and 2022. Average suspended sediment flux (e), total precipitation (f), and total rainfall (g) during hydrological events associated with extreme precipitation events, non-extreme precipitation ($P_{\mathrm{mean}} > 0.1 \mathrm{~mm~h}^{-1}$), and no precipitation ($P_{\mathrm{mean}} \leq 0.1 \mathrm{~mm~h}^{-1}$) show significant differences in their magnitude (two-sample Kolmogorov-Smirnov test with 5 % significance level, see Hodges, 1958). The number of hydrological events in each category with valid data is shown above each boxplot.

throughout the 17 years (Fig. 7b). In Vent-Rofental, the contribution of hydrological events associated with non-extreme or no precipitation decreased until 2021, though abruptly increased in 2022 (Fig. 7d).



There is a significant difference between the $\mathrm{SSF_{mean}}$, $P_{\mathrm{tot}}$, and $\mathrm{RF_{tot}}$ of hydrological events associated with extreme,
non-extreme and no precipitation (Fig. 7e-g). Those with extreme precipitation have the highest magnitudes, followed by non-
extreme and no precipitation. On average, 23 % of annual suspended sediments were exported during extreme precipitation
events, compared to 37 % during non-extreme precipitation and 40 % during hydrological events without precipitation. The
discrepancy between the magnitude and annual contribution of each category is due to the differing event frequencies (Fig. 7e-
g). Events influenced by extreme precipitation are rarer, thus despite their high $\mathrm{SSF_{mean}}$, their contribution to annual SSY is
lower than the other two categories.

### 4.5 Fraction of precipitation-driven suspended sediment spikes

The results of the inverse analysis showed that only 10 % of the more than 1000 suspended sediment spike events, i.e. hydro-
logical events with $\mathrm{SSC_{max}} > P_{90}(\mathrm{SSC}_t)$, only about 10% were associated with extreme precipitation (Fig. 8). About a third
were associated with some, but not extreme, precipitation (Fig. 8). More than half of the suspended sediment spikes in both
catchments were not associated with any precipitation. The pattern was nearly identical for Tumpen-Ötztal and Vent-Rofental.



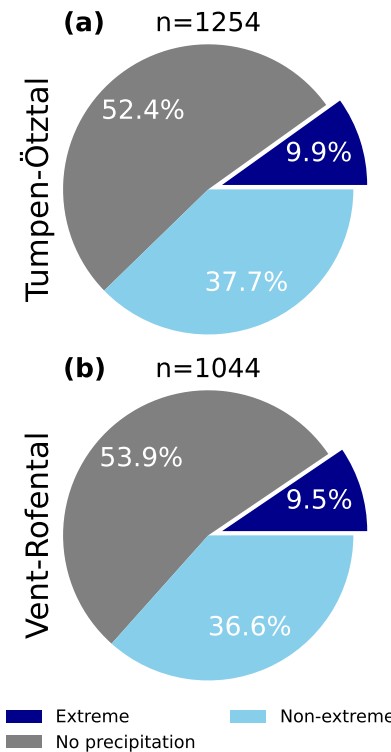

**Figure 8.** Number of suspended sediment spikes (hydrological events with $\mathrm{SSC_{max}} > P_{90}(\mathrm{SSC}_t)$) associated with extreme precipitation events, non-extreme precipitation ($P_\mathrm{mean} > 0.1 \ \mathrm{mm\,h^{-1}}$), and no precipitation ($P_\mathrm{mean} \leq 0.1 \ \mathrm{mm\,h^{-1}}$) in Tumpen-Ötztal (a) and Vent-Rofental (b).

## 5 Discussion

### 5.1 Uncertainty analysis

#### 5.1.1 INCA uncertainty and detection of precipitation extremes

Our uncertainty analysis found that INCA tends to somewhat overestimate both the occurrence and amount of precipitation,
355 with increasing inaccuracies with elevation. Using a very dense rain-gauge network in southeastern Austria, Ghaemi et al. (2021) also found both over- and underestimation in space of INCA annual and heavy precipitation. They also found higher errors in 2012-2014, which they attribute to the installation of a new radar, and reported an improvement after 2015. We find that INCA uncertainty is higher for the first 13 years of the study period with a marked reduction in 2017, which may also be related to the incorporation of the new radar. Furthermore, INCA tends to underestimate precipitation at the highest-elevation
360 stations in the south-western part of the catchment, where the terrain is enclosed by mountain peaks over 3000 m asl. This underestimation may therefore be caused by radar beam shielding (Germann et al., 2006).



The changes in INCA uncertainties over time may affect the number of events detected by our threshold-based approach. However, given that before 2017 INCA had higher annual mean errors (i.e. positive bias) and a greater tendency to overpredict heavy precipitation days, it is more likely that errors in INCA inflated the event numbers in the first 13 years of the study period. Hence, we assume that the significantly positive trend in annual event numbers is not affected by the errors in INCA.

Except for the three detected events that were in fact data artefacts, we found no indications that the errors in INCA produced further false positives. All other 166 events were verified as precipitation events recorded at weather stations. Despite uncertainties in the estimated precipitation quantities, as our detection routine relies on thresholds estimated from the INCA data itself and the spatial extent of precipitation, we believe that the uncertainties in INCA hardly affect our detection routine.

Moreover, a combined gridded precipitation product such as INCA has clear advantages: First, complete and consistent spatial coverage with information on areas not covered by weather stations due to the inclusion of radar-based remotely sensed precipitation. Second, a spatial resolution of 1 km and temporal resolution of 1 h, which resolves grid-scale sub-daily extremes like convective storms (e.g. event 2022-d, Fig. 4a). The spatio-temporal characteristics of such events could not have been detected with weather stations alone and their rainfall intensities would likely have been underestimated (Schroeer et al., 2018). Furthermore, geomorphological variables are sensitive to small-scale changes in rainfall spatial structure (Peleg et al., 2020), which can only be captured by distributed, high-resolution data. Finally, using a dataset with consistent spatio-temporal extent and resolution has advantages over using stations with varying temporal coverage and spatial representativeness. Hence, the detection and characterisation of precipitation extremes at multiple spatial scales would have been impossible from station data alone and would have missed key precipitation characteristics that are relevant for rainfall erosion and sediment transport processes.

### 5.1.2 Methodological uncertainties in event detection

Our assumption of stationarity in the precipitation time series has important implications, as it may introduce uncertainty into the thresholds used for event detection. By assuming stationarity, we neglect the possibility that the underlying precipitation distribution is shifting (or scaling) over time due to a covariate, such as global mean temperature. Both observation-based and model-based studies suggest that heavy precipitation events in the European Alps are likely to increase with climate warming (Giorgi et al., 2016; Molnar et al., 2015; Dallan et al., 2024; Vergara-Temprado et al., 2021).

Consequently, non-stationarity in extreme precipitation in our study area is likely. If true, this assumption introduces uncertainty into the estimation of the d-GEV. However, two aspects lead us to believe that this is of less concern. First, with a relatively short study period of 21 years, shifts in extreme values will only have limited influence. Second, our primary interest lies in understanding how shifts in extreme precipitation over time affect sediment transport. Many geomorphic processes, such as erosion and sediment transport, respond only when certain thresholds are exceeded. Consequently, adjusting detection thresholds to account for temporal shifts in extreme precipitation would not necessarily reflect the physical precipitation-driven hydro-geomorphic response in the catchment. Furthermore, the fact that we use multi-scale detection and 0.8 exceedence-probability to define detection thresholds means that we are quite generous with our definition of "extreme".





## 5.2 Drivers of fluvial sediment export during precipitation extremes

### 5.2.1 Importance of distinguishing liquid and solid precipitation

We find a close positive association between rainfall intensities of extreme precipitation events and their suspended sediment yields, fluxes and peak concentrations. Precipitation phase significantly influences runoff generation, which may explain why we find that precipitation alone has a weaker association with suspended sediment responses compared to liquid precipitation. In the steep, high-altitude terrain of the Alps, precipitation during summer can still fall as snow at higher catchment elevations. Distinguishing between rain and snow with elevation is already emphasised in hydrological modelling and flood forecasting (Harpold et al., 2017). Moreover, in high-elevation regions of the Northern Hemisphere, increases in rainfall extremes with climate warming is amplified due to a warming-induced shift from snow to rain, making them hotspots with increasing risk of extreme-rainfall-related hazards (Ombadi et al., 2023). Our results highlight that precipitation phase is an important factor to quantify and consider when studying sediment transport during extreme precipitation events in Alpine areas.

### 5.2.2 Sub-daily precipitation extremes and sediment dynamics

Our analysis shows that, at lower rainfall intensities, sub-daily precipitation extremes generate higher suspended sediment yields and fluxes compared to long-duration ones. Sub-daily extremes affect a smaller catchment area (Fig. 4e) meaning that the precipitation will be more localised resulting in higher local rainfall intensities than the catchment-averaged values. Since erosion rates are sensitive to increases in rainfall intensity (Shen et al., 2016; Nearing et al., 2005; Berger et al., 2010), sub-daily extremes can induce locally high erosion rates (Battista et al., 2020) producing high event yields even at lower rainfall totals (Fig. 6c). Furthermore, sub-daily extremes are primarily observed during summer, when convective storms dominate. Convective summer storms generate localised high-intensity rainfall, which preferentially drives hillslope erosion (Kampf et al., 2016) and increases sediment yields, particularly through rill erosion (Shen et al., 2016). Higher intensity storms are also associated with higher functional sediment connectivity, as shown by (Scorpio et al., 2022), allowing sediment mobilised on hillslopes to efficiently reach channels.

During sub-daily precipitation extremes, overland flow is most likely generated mainly due to infiltration excess where the rainfall intensity exceeds the infiltration capacity of the surface. Runoff generation due to infiltration excess is highly sensitive to temporal variations in rainfall intensity (Bronstert and Bárdossy, 2003). In particular, short bursts of high-intensity rainfall significantly influence infiltration processes and rainfall-driven erosion (Dunkerley, 2019, 2021b; Bronstert and Bárdossy, 2003). However, such high-intensity rain bursts often cannot be resolved at hourly resolution (Dunkerley, 2019, 2021b). Thus we are unable to verify whether such rain bursts occur during sub-daily extremes in our analysis. Nonetheless, we assume that sub-daily extremes with their more localised precipitation exhibit greater temporal intensity variation than long-duration events, making them more likely to generate overland flow and enhance erosion. As Dunkerley (2021a) notes: "Low-intensity rain may be readily absorbed until rainfall ends or until saturation occurs, whilst high-intensity rain may quickly exceed the soil infiltrability, resulting in surface ponding and integrated downslope overland flow."





Rainfall-driven geomorphic processes such as erosion and sediment transport exhibit threshold behaviours (Peleg et al., 2020). The localised high intensities of sub-daily extremes can more readily exceed these thresholds, activating erosion processes and generating significant sediment fluxes. Another relevant process which is also highly threshold-dependent is mass wasting. Certain sub-daily events in our catalogue stand out for their unusually high SSC peaks despite relatively low rainfall intensities (Fig.6i-j) and most of these were also classified as grid-scale extremes (Fig. A4i-j). One such event is 2022-d (Fig. 4a), which triggered more than 150 debris flows (Himmelstoss et al., 2024; Rom et al., 2023). We hypothesise that these events represent highly localized precipitation triggering intense erosion, sediment mobilisation, and possibly mass wasting, which deposit large sediment pulses into the main channel and result in exceptional SSC peaks at the outlet.

### 5.2.3 Sediment response to long-duration precipitation extremes

Long-duration precipitation extremes distribute rainfall over extended periods, which generally result in lower average rainfall intensities compared to sub-daily extremes (Fig. 6). When rainfall intensity remains below the soil's infiltration capacity, surface runoff and associated erosion is less widespread, leading to lower sediment yields. As such, many long-duration extremes may lack the rainfall intensities required to trigger hillslope erosion or the functional connectivity to transport sediments to the streams. However, their high rainfall totals (Fig. 4g) can generate high streamflow levels through rapid runoff concentration, which rather results in in-channel erosion (Kampf et al., 2016; Scorpio et al., 2022).

Conversely, as long-duration events progress and the ground becomes increasingly saturated, runoff generation through saturation excess becomes more likely. Saturated conditions can destabilise soils and increase their erodibility (Amézketa, 1999) and saturation excess overland flow can still cause rill and gully erosion (Kirkbride and Reeves, 1993). This may explain why some long-duration extremes with high rainfall intensities produce the highest suspended sediment yields (SSY; Fig. 6a-b) and why they increase more sharply with rainfall intensities compared to sub-daily extremes. Events characterized by low initial rainfall intensity with a later intensification are likely to play a role in this effect, as these "late-peak" events are associated with higher runoff ratios and peak overland flow rates (Dunkerley, 2021a).

Despite lower, more distributed rainfall intensities, long-duration extremes can exhibit complex intra-event rainfall dynamics, as they may include short bursts of high intra-event rainfall rates. Dunkerley (2021a) demonstrated this in Australian data, showing that longer events can produce elevated 30-minute intensity peaks values despite their lower overall intensities. In our results, the subset of long-duration extremes that also contained sub-daily extreme durations exhibited markedly higher suspended sediment yields, fluxes, and peak concentrations as well as rainfall intensities and amounts compared to those that were only extreme at durations $\geq 24$ h (Fig. 9). These *long-mixed* extremes also had markedly higher rainfall totals and SSY compared to sub-daily extremes (Fig. 9). This suggest that high-intensity rain bursts nested within long-duration extremes is a key driver of exported sediment mass. This highlights the benefit of our multi-scale detection approach, as the identification of multiple extreme durations allows us to classify such events with complex intra-event rainfall dynamics.

Another key feature of long-duration extremes is their large spatial extent, which typically affects a broader catchment area compared to sub-daily extremes (Fig. 4e). Consequently, erosion processes can occur over a larger area and for a longer time. Thus even if the erosion is less intense compared to sub-daily extremes, it can result in greater total sediment volumes overall.



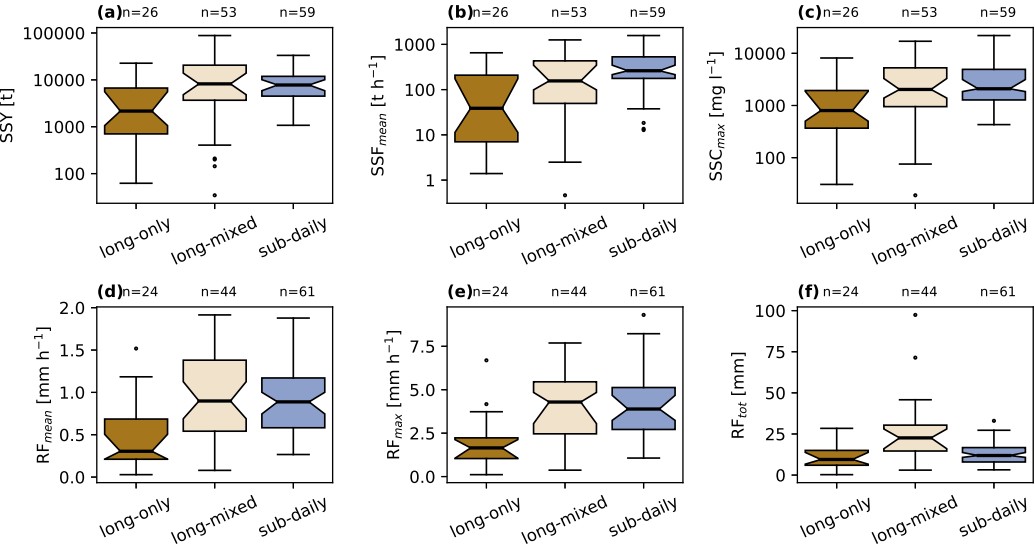

**Figure 9.** Sediment response variables (a-c) and rainfall characteristics (d-f) of precipitation extremes. Long-duration extremes are divided into two sub-classes: "long-only" events where all extreme durations are $\geq 24$ h, and "long-mixed" events which also have some extreme durations at the sub-daily scale. See Fig. 4 for details on boxplot configuration.

Thus, while long-duration extremes may lack the intense, localized rainfall bursts that drive erosion in sub-daily events, their broader spatial and longer lasting impact can produce substantial sediment fluxes at the catchment scale.

## 5.3 Trends in extreme-precipitation-driven suspended sediment yields

Over the study period, we observe an increase in the fraction of annual SSY exported during precipitation extremes. Further-
more, the number of precipitation extremes significantly increased. As discussed above, uncertainties in the INCA dataset do not appear to significantly affect the detected number of extreme events. Lower numbers of precipitation extremes in the early study period likely reflect drier-than-average years, such as 2004 and 2006 (Gattermayr et al., 2004, 2006), dominated by isolated convective storms. The significantly increasing trend in the number of precipitation extremes is therefore more likely due to an overall increase in precipitation intensity leading more events to exceed the extreme thresholds. While we find no
indications of the precipitation or rainfall intensity of extremes significantly increasing over the study period, this may be due to the high inter-variability of precipitation and the relatively short period we are considering. May-October precipitation in Ötztal calculated from the SPARTACUS dataset (GeoSphere Austria, 2024c), consisting of spatially interpolated station data (see Hiebl and Frei, 2018), has significantly increased with 2.6 mm per year since 1961 (MK test with 5% significance level). Thus we can at least say that the total precipitation amount during the extended summer season increased during our study pe-
riod. As extreme precipitation-driven transport events have significantly higher suspended sediment fluxes (Fig. 7e), a pattern





common in mountainous areas (e.g. Li et al., 2021b; Rainato et al., 2021; Wulf et al., 2012; Scorpio et al., 2022; van Hamel et al., 2025), the increasing number of extreme precipitation events means that the fraction of annual SSY also increases.

In years with a high frequency of precipitation extremes, this effect is especially noticeable. Examples are 2010 and 2020, years in which around 40% of annual SSY were associated with extreme-precipitation (Fig.7a). In 2020, extreme precipitation
events contributed significantly to annual sediment export. For instance, during event 2020-k, persistent rainfall triggered a debris flow in the proglacial area of the Hintereisferner glacier (in Vent-Rofental). A thunderstorm in August (2020-j) and a cold front in October (2020-n) caused additional flooding and mass wasting. Together these three events contributed 124 t km$^{-2}$ or 25 % of the annual SSY of 2020. The remaining 13 events contributed 15%. The year 2020 illustrates the cumulative effect of both moderate and severe storms on sediment yields (Wischmeier and Smith, 1978). Notably, the years 2013 and 2019 had
the same number of extreme precipitation events but did not contribute comparable amounts or fractions of annual SSY. This highlights that factors such as annual sediment production and availability play an important role in determining the fraction of extreme-precipitation-driven sediment exports, in addition to the annual frequency and intensity of precipitation extremes.

Annual SSYs have remained stable over the study period in Tumpen-Ötztal but show a gradual decline in Vent-Rofental (Fig.7), excluding extreme melt years such as 2022 (Bayerische Akademie der Wissenschaften, 2025; Voordendag et al., 2023).
In Vent-Rofental, peak sediment may already have been passed (Schmidt, 2023), with annual SSY now steadily decreasing as glacier retreat reduces subglacial sediment supply, a trend projected to persist over the long term (Schmidt et al., 2024). In Tumpen-Ötztal, sediment exported during non-extreme or precipitation-free events showed a non-significant decline (Fig.7b), suggesting that increases in extreme-precipitation-driven sediment transport at larger scales may partially offset reductions in glacier-driven transport. These contrasting trends highlight the influence of different spatial scales of paraglacial adjustments,
as the timing of peak sediment and the duration of sediment reworking vary with catchment size and glacier cover (Ballantyne, 2002).

Despite the increasing prominence of extreme precipitation, only 10% of extreme suspended sediment spikes in Ötztal can be directly attributed to these events, with nearly half occurring without precipitation (Fig. 8). This is consistent with findings showing that high SSC peaks in Alpine catchments can be generated by rainfall, glacier melt, snowmelt and combinations
thereof (Skålevåg et al., 2024; van Hamel et al., 2025). Melt processes, due to their frequency, remain the dominant driver of annual SSY (Schmidt et al., 2022; Skålevåg et al., 2024), with the most extreme sediment discharge often arising from a combination of melt- and precipitation-driven processes (Skålevåg et al., 2024; van Hamel et al., 2025). Thus extreme-precipitation-driven sediment transport in this study likely have glacial contributions to sediment transport, especially during summer, when subglacial sediment discharge is elevated.

While glacial influences on sediment transport are declining, the projected increase in extreme precipitation, particularly at sub-daily scales, remains a critical factor. Conceptual models suggest that rainfall-driven sediment transport will dominate post-deglaciation sediment yield levels (Zhang et al., 2022b). Convective summer precipitation is expected to intensify in the Alps (Giorgi et al., 2016), with an ensemble of 1-km convection-permitting climate models projecting a 20–38% increase in sub-daily (1–24 h) extremes in the Eastern Alps, including Tumpen-Ötztal (Dallan et al., 2024). Our results demonstrate that
sub-daily precipitation extremes are particularly effective at generating high suspended sediment loads. This is amplified by



the abundant unconsolidated sediments in high-elevation landscapes, a legacy of deglaciation, which will remain available for transport until these areas stabilize (Musso et al., 2020; Ballantyne, 2002). While vegetation and soil development will eventually promote stabilisation (Klaar et al., 2015), these processes are temporally variable (Bayle et al., 2023) and can be disrupted by geomorphic disturbances such as rainfall erosion and fluvial reworking of proglacial deposits (Moreau et al.,

2008). These dynamics suggest that future sediment regimes in high-elevation catchments like the Ötztal will become flashier, characterised by more frequent and intense rainfall-driven events, but with overall lower annual yields.

## 6 Conclusions

This study provides new insights into the interplay between extreme precipitation, sediment transport, and paraglacial adjustments in high-Alpine environments.

Sub-daily precipitation extremes, driven primarily by convective summer storms, are particularly effective at mobilizing sediment due to their localized and intense rainfall, which exceeds erosion and runoff thresholds. Long-duration precipitation extremes, while less intense, affect larger catchment areas and sustain sediment transport over longer timescales, especially when they include sub-daily rainfall bursts.

We observed a significant increase in the frequency of extreme precipitation events and their contribution to annual sus-

pended sediment yields (SSY) over the study period. While we find robust evidence of an increasing trend in extreme precipitation events, uncertainties in the INCA dataset, such as its tendency to overestimate heavy precipitation, particularly at higher elevations, may partially influence this trend. However, given the reduction in INCA errors post-2017 and the strong alignment between detected extremes and station data, these uncertainties likely have a limited effect on our overall findings.

Despite the increases in extreme-precipitation-driven sediment transport, annual SSY has remained stable in Tumpen-Ötztal

but declined in Vent-Rofental, where reduced sediment availability due to glacier retreat appears to drive long-term declines. This suggests that increases in extreme-precipitation-driven sediment transport at larger scales may partially offset reductions in glacial-driven transport in less glaciated catchments. However, sediment transport during extreme events is highly dependent on sediment availability, highlighting the role of glacial and paraglacial dynamics in controlling sediment fluxes.

Projections indicate that extreme precipitation, particularly at sub-daily scales, will intensify in the future due to climate

warming. This, combined with the abundant unconsolidated sediment in deglaciating landscapes, suggests that Alpine catchments like the Ötztal will experience flashier sediment regimes characterized by more frequent and intense rainfall-driven events. However, long-term stabilization of these landscapes will depend on changes in vegetation cover and soil development, processes that are temporally variable and vulnerable to disturbance.

Overall, while the intensification of extreme precipitation events is expected to increase the frequency of sediment transport

events, long-term reductions in glacial sediment supply will likely result in declining annual sediment yields. These findings emphasise the need for continued monitoring and refined modelling of sediment transport dynamics under changing climatic and geomorphic conditions in Alpine environments.



*Code and data availability.* The HISTALP dataset with monthly precipitation amounts (GeoSphere Austria, 2020), INCA hourly precipitation and temperature grids (GeoSphere Austria, 2024a), and hourly precipitation data from GeoSphere Austria's weather stations (GeoSphere

Austria, 2024b) are available for download at the GeoSphere Austria Data Hub, https://data.hub.geosphere.at. Daily precipitation data from the 15 rain gauges operated by the Hydrographic Service of Tyrol (HD-Tirol) were downloaded from https://ehyd.gv.at/ (BML, 2024). Precipitation data from Hintere-Fundusalm with 15-min resolution was provided by HD-Tirol upon request. Raw precipitation data from the Vent, Hochebenkar, and Station Hintereis weather stations were downloaded from ACINN (2024). Precipitation data in 10-min resolution from weather stations Latschbloder, Bella Vista, and Proviantdepot operated by Department of Geography - University of Innsbruck (2024).

Processed data, results, and code can be found here: https://doi.org/10.5281/zenodo.16571983.

**Appendix A**



**Table A1.** Weather stations in and around Ötztal with precipitation observations used to evaluate the uncertainty of INCA gridded precipitation. Most stations are operated by GeoSphere Austria (GSA) or the Hydrographic Service of Tyrol (HD-Tirol) with additional stations operated by the Department of Geography (UIBK-GEOG) and the Department of Atmospheric and Cryospheric Sciences (ACINN) at the University of Innsbruck, and the Bavarian Academy of Sciences (BADW). The original data had temporal resolutions ranging from 1-min to daily and has varying temporal extents. For an overview of available data during the study period see Fig. A1.

| Station | | Location | | | Temporal | |
|---|---|---|---|---|---|---|
| Name | Operator | Latitude | Longitude | Altitude | Resolution | Extent |
| Stams | HD-Tirol | 47.2744 | 10.98528 | 711 | daily | 1971-2021 |
| Oetz | HD-Tirol | 47.2058 | 10.88611 | 760 | daily | 1971-2021 |
| Imst-Oberstadt | HD-Tirol | 47.2489 | 10.74111 | 860 | daily | 1971-2021 |
| Ried-im-Oberinntal | HD-Tirol | 47.0575 | 10.66056 | 895 | daily | 1971-2021 |
| Jerzens-Ritzenried | HD-Tirol | 47.1225 | 10.78194 | 1120 | daily | 1971-2021 |
| Längenfeld | HD-Tirol | 47.0763 | 10.97000 | 1180 | daily | 1971-2021 |
| Gries-im-Sellrain | HD-Tirol | 47.1886 | 11.15250 | 1227 | daily | 1971-2021 |
| Kaunertal-Vergötschen | HD-Tirol | 47.0431 | 10.75306 | 1269 | daily | 1971-2021 |
| St.Leonhard-i.P. | HD-Tirol | 47.0761 | 10.83889 | 1329 | daily | 1971-2021 |
| Sölden | HD-Tirol | 46.9850 | 11.01472 | 1332 | daily | 1971-2021 |
| Ladis-Neuegg | HD-Tirol | 47.0972 | 10.64889 | 1426 | daily | 1971-2021 |
| Gries | HD-Tirol | 47.0700 | 11.02667 | 1590 | daily | 1998-2021 |
| Plangeroß | HD-Tirol | 46.9875 | 10.86750 | 1605 | daily | 1971-2021 |
| Ochsengarten-Obergut | HD-Tirol | 47.2300 | 10.91861 | 1695 | daily | 1971-2021 |
| Hintere-Fundusalm | HD-Tirol | 47.1028 | 10.88611 | 1960 | 15-min | 2020-2023 |
| Dresdner-Hütte | HD-Tirol | 46.9975 | 11.14000 | 2290 | daily | 1979-2021 |
| Innsbruck-Flugplatz | GSA | 47.2600 | 11.35667 | 578 | hourly | 1992-2024 |
| Innsbruck-Univ- | GSA | 47.2600 | 11.38417 | 578 | hourly | 1986-2024 |
| Haiming | GSA | 47.2597 | 10.88944 | 659 | hourly | 2007-2024 |
| Imst | GSA | 47.2369 | 10.74222 | 773 | hourly | 2007-2024 |
| Landeck | GSA | 47.1403 | 10.56361 | 796 | hourly | 1993-2024 |
| Neustift-Milders | GSA | 47.1028 | 11.29195 | 1007 | hourly | 2004-2024 |
| Umhausen | GSA | 47.1392 | 10.92889 | 1035 | hourly | 2003-2024 |
| St-Leonhard-Pitztal | GSA | 47.0272 | 10.86556 | 1454 | hourly | 2007-2024 |
| Obergurgl | GSA | 46.8670 | 11.02445 | 1941 | hourly | 1999-2024 |
| Pitztaler-Gletscher | GSA | 46.9270 | 10.87917 | 2863 | hourly | 1994-2024 |
| Vent | ACInn | 46.8583 | 10.91250 | 1890 | daily | 1935-2016 |
| | | 46.7989 | 10.91276 | 1908 | 1-min | 2015-2024 |
| Hochebenkar | ACInn | 46.8370 | 11.00822 | 2565 | 10-min | 2012-2014 |
| Station Hintereis | ACInn | 46.7989 | 10.76037 | 3031 | 1-min | 2020-2023 |
| Proviantdepot | UIBK-GEOG | 46.8595 | 10.82407 | 2737 | 10-min | 2019-2023 |
| Bella Vista | UIBK-GEOG | 46.7828 | 10.79138 | 2805 | 10-min | 2015-2023 |
| Latschbloder | UIBK-GEOG | 46.8011 | 10.80659 | 2919 | 10-min | 2013-2023 |



**Figure A1.** Overview of available (blue) and missing data (white) for weather station observed precipitation at hourly (a) and daily (b) resolution during the study period (2004–2024). Stations are sorted lowest (bottom) to highest (top) elevation.



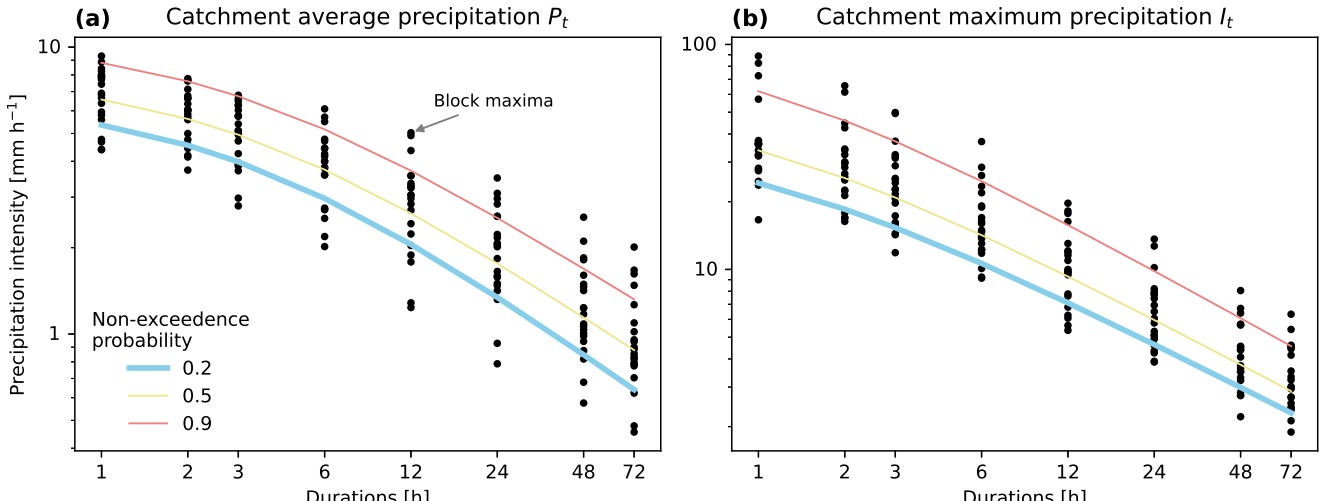

**Figure A2.** Intensity-duration-frequency (IDF) curves for catchment-averaged precipitation $P_t$ and grid-scale maximum precipitation $I_t$. Black dots show annual May-October block maxima for each duration. The 0.2 non-exceedence probability (blue line) was the threshold used to detect extreme precipitation events in $P_t$ and $I_t$.







**Figure A3.** Uncertainty analysis of daily and hourly INCA precipitation when compared with weather station data. Each metric, the mean error (ME), root mean squared error (RMSE), precipitation occurrence accuracy (Acc), and precipitation occurrence frequency bias (FB) is plotted against the altitude of the station. Dot size is proportional to the number of time steps with valid INCA and station value pairs.





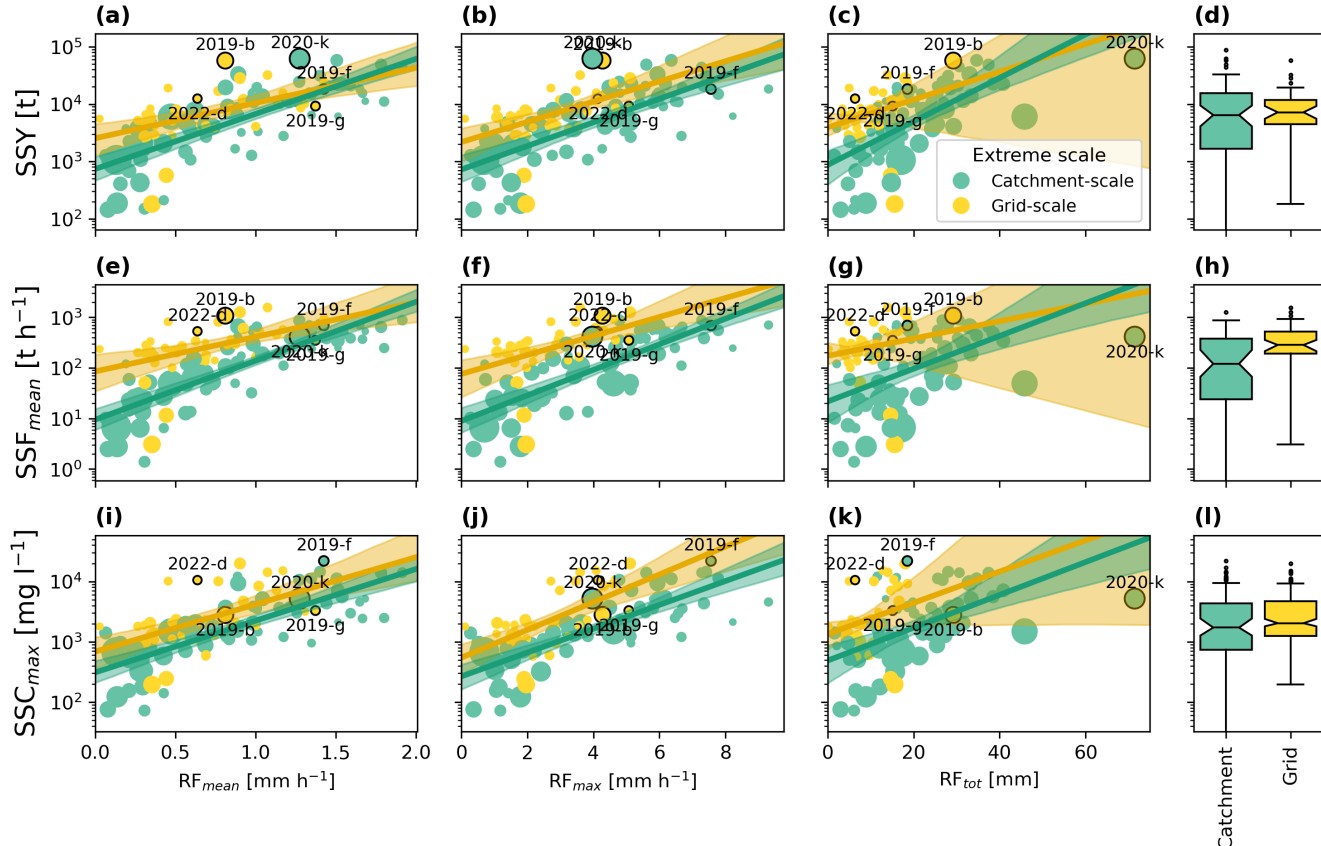

**Figure A4.** Suspended sediment response to precipitation extremes in terms of their mean intensity of catchment-averaged rainfall $RF_{mean}$(a,e,i), maximum intensity of catchment-averaged rainfall $RF_{max}$ (b,f,j), and total cumulative catchment-averaged rainfall $RF_{tot}$ (c,g,k). The increase in sediment response variables with rainfall intensity and totals is similar for both grid-scale extremes and catchment-scale extremes. Lines are means from linear regression, and shaded areas are 95% bootstrap confidence intervals based on 1000 randomisations. Labelled events are the same as in Fig. 4. See Fig. 4 for details on boxplot configuration.



*Acknowledgements.* The authors would like to acknowledge Maik Heistermann for his advice on how to analyse precipitation uncertainty and for the idea to use two time series for the detection, i.e. both catchment-averaged and grid-scale maximum precipitation. The research presented in this article was conducted within the research training group "Natural Hazards and Risks in a Changing World" (NatRiskChange) funded by the Deutsche Forschungsgemeinschaft (DFG; GRK 2043/2). During the writing and revision of the manuscript, ChatGPT assisted in editing existing portions of text.



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
