# Peer review of "Linking extreme rainfall to suspended sediment fluxes in a deglaciating Alpine catchment"

_EGUsphere, 2025_

## Referee Comment (RC2)

**General comments**

This manuscript addresses the critical and timely topic of suspended sediment dynamics in a rapidly deglaciating Alpine catchment under the influence of increasing extreme rainfall events. This work is highly relevant to the scope of Hydrology and Earth System Sciences (HESS) as it provides quantitative insights into how climatic shifts are altering hydro-geomorphic processes in high-mountain environments. The study is well-conducted and presents a thorough analysis linking extreme precipitation characteristics to suspended sediment fluxes in the Tumpen-Ötztal and Vent-Rofental catchments in Austria. The methods are robust and clearly described, and the resulting data analysis is systematic. The manuscript is well-written, and the discussion is generally comprehensive. Overall, I recommend publication of the manuscript following minor revisions. Some comments are listed below.

**Specific Comments**

Antecedent Catchment Conditions. The discussion (lines 484–486) highlights the role of sediment availability for interpreting the year-to-year variability in SSY. In my opinion, authors should also consider the influence of antecedent catchment conditions, specifically factors like soil moisture, which might impact both overland flow generation and soil erosivity, the sequence of events, which dictates the depletion of readily available in-channel and hillslope sediment supply, and the presence/absence of snow cover, which might modulate the rainfall-SSY relationship. An event-based analysis of few targeted events could provide insights on these factors.

Snow and Ice Melt Processes. In general, little space is given in the discussion on the impact of snow and ice melt processes on sediment transport. I suggest the authors expand this discussion to address the temporal dynamics and potential overlap with rainfall extremes. Specifically: When does the peak ice melt happen? Is it overlapping with the period characterized by the highest short-duration convective rainfall? Could this interaction explain the higher SSFmean observed for short-duration events? Furthermore, describing a typical pattern of snow cover duration in the catchment would add context. Again, by analyzing a few events with different characteristics, as they did for 2020, the authors would be able to incorporate these key cryospheric processes more fully into the discussion.

**Event classification.** I have a clarifying question regarding the event classification described in Lines 224–226. Does this methodology imply that a genuine long-duration event could be identified or partially characterized as a sub-daily extreme if it contains a single, very intense sub-daily peak? I am not sure I understood this and I wonder how potential mis-classification might influence the results showed in Figure 6 as well as the discussion in 5.2.2. and 5.2.3.

Increasing frequency of extreme precipitation – Stations. The finding of an increasing trend in the frequency of extremes derived from the INCA product is central to the study's context. Have the authors checked if a similar increasing trend is observable in the precipitation station data used for INCA development? Recognizing the already thorough nature of the analysis, I suggest the authors check if a similar increasing trend is observable in some targeted stations. This would enhance the robustness of the signal by ruling out the possibility of the trend being an artifact of the gridded product or its calibration process.

**Technical Corrections**

Figure 2. The font size is rather small. The figure could be a bit bigger.

Line 100-101, 108-109: In the sentence "The accuracy of INCA estimates can vary, particularly in complex terrain, with an average error of 50-100% in the 15-minute precipitation grids and 1.0 to 1.5 °C in the temperature grids (Haiden et al., 2011).", the 15-minutes precipitation grids confuses because in line 100-101 the authors describe the INCA datasets as "... hourly 1-km grids for all of Austria.". Perhaps just add "... and sub-hourly ..." to the sentence in line 100-101.

Line 115-116: Please, provide a brief (few words) explanation of the rainfall/snowfall separation method used in openAMUNDSEN.

Line 120-123: As I understand here, you used hourly, 1-km grids, precipitation for the period 2004-2024, and rainfall for the period 2011-2024. Correct? Please, clarify.

Line 125. Figure 1 to Figure 1b.

Line 190: Please improve clarity by changing "Detection thresholds, u, for each ..." to "Detection thresholds, u, for each DURATION d and spatial scale (i.e. GRID-SCALE It or CATCHMENT-AVERAGED Pt).

Line 210: I believe I understood what you did, but could you please write this iterative merging in a clearer way?

Line 265: Since you are talking about events with SSC larger than the 90th percentile, I find P90(SSCt) a confusing definition and would change it to SSC90.

Figure 8. Font size rather too large. I suggest being more precise in the legend: from "extreme" to "extreme precipitation", from "non-extreme" to "non-extreme precipitation".

---

## Author Comment (AC1)

**Anonymous Referee #1 from 20 Oct 2025**

https://doi.org/10.5194/egusphere-2025-3683-RC1

**Comments**

I should note that my background is not in the kind of precipitation analyses presented in the paper but rather sediment sources and transport in glacial and paraglacial systems. This means that my review of the paper is only partial.

Overall, this paper is worth publishing in my view. It is extremely well-written and well-presented. It has an importance as whilst it builds upon similar papers from authors of this one, and with overlapping geographical foci, the topic is important as there is an ongoing debate over how the transition from glacier-melt dominated to rainfall-dominated catchments impacts suspended sediment yield from deglaciating basins. The analysis is largely very well done (I make some more minor comments below). On this basis I would hope that the paper can be published after some revision.

That said, I do think the authors need to be much more careful in how they present and interpret this work and I would recommend that they make some revisions.

**1.** The general positioning of the paper is stronger on the hydrology than on the glaciology. There is actually a very established literature on how glaciated catchments produce suspended sediment and the processes that drive it (e.g. the work of Darrel Swift, Angela Gurnell, David Hannah) and I think this should be brought into the paper. Note David Hannah's work is also important for hydrograph analysis in this kind of setting. This is not least because you cannot assume that extreme precipitation and melt-driven suspended sediment production events are entirely independent. Some of the previous work by the authors recognises this and the point is made at l502. It is quite possible that your extreme rainfall events do not change erosion but rather increase sourcing of subglacial suspended sediment. We know that surcharge of subglacial drainage systems by water, whether melt, precipitation or precipitation-indeed melt, leads to very major suspended sediment yield, especially in "spring events". These don't just occur in spring but can occur at any time and they likely interact with other variables such as snow cover on the glacier. The discussion that then follows from around l410+ here appears to be highly speculative as we have nothing on this kind of driver but a lot on classical kinds of suspended sediment sources associated with very different kinds of environments.

> *Reply: The referee makes an excellent point about precipitation and subglacial sourcing. We now introduce the melt component to the extremes much earlier in the manuscript, and also include this in the interpretation of processes responsible for the detected relationships and patterns.*

**2.** Part of the problem here is that with two gauging station records of this kind, separating out individual and joint effects is not easy, and not least because (see below), your flux and yield data are collected from discharge which in turn will be a function of precipitation; so when you correlate precipitations variables with flux and yield data you have an inevitable correlation. The authors might have been much better to do a more event-based analysis than they have done.

> *Reply:*

*Regarding correlation between precipitation characteristics and sediment response variables*

*While we do include correlations between heavy precipitation event characteristics and corresponding suspended sediment response (e.g. Table 2), we by no means argue that the fact that rainfall characteristics and sediment flux is correlated is the main result. The two main findings we wish to highlight is:*

*1. Differing responses in fluvial sediment transport during sub-daily and long-duration extremes.*

*2. The role that heavy precipitation events play in the fluvial sediment transport regime and whether their influence in changing.*

*We believe this is also reflected in this taking up most of the discussion. However, as the reviewer notes, we should be more careful in our interpretation and discussion of processes responsible for the detected relationships and patterns. We will rework the discussion to include a more nuanced interpretation of processes.*

*Regarding the comment to do a "more event-based analysis"*

*The referee does not specify what type of event-based analysis they want to see.*

*First, we would argue that our analysis is already "event-based" in the sense that the basis for our analysis is to partition the data into events. An analysis could for example be based on daily data where the whole sediment transport season is analysed.*

*Second, there is actually more event-based analysis and evaluations "under the hood" of our analysis presented in the manuscript. The figure below is an example of a "dashboard" created for each heavy precipitation event to examine the precipitation pattern, include differences between the gridded precipitation product and stations, and show the hydrograph and sedigraph at the two gauging stations.*

*We propose to expand this dashboard figure to include the hysteresis pattern, and, if feasible, information of snow conditions and melt-potential, e.g. positive degree days, which was also requested by another referee. The figures for all events will be added to the supplementary materials. We do however think, that included too much discussion of single events would dilute the main findings of the paper as highlighted above. We see the main contribution of our paper to be a contribution to the "ongoing debate over how the transition from glacier-melt dominated to rainfall-dominated catchments impacts suspended sediment yield from deglaciating basins" as the referee very nicely puts it.*

[Figure]

**3.** The same problem under (1) is then reflected in the literature, how the paper is both set up and discussed. I am not sure how much time the authors have actually spent in this kind of river basin during extreme events, but I am concerned to see a number of papers developed for suspended sediment / soil erosion in very different environments being transferred uncritically into this one. Is Wischmeier and Smith (1978) really relevant to this case? Dunkerley's excellent work from Australia? Processes are invoked to explain results (e.g. soil erosion due to overland flow) when field measurements of such processes question whether they really occur in at least some of the environments described here. Specifically, it is really rare to see classic overland flow in these kinds of catchments in these kinds of environments unless there is some interaction with frozen ground. The unconsolidated sediments you note on l510+ have exceptionally high vertical infiltration rates unless they contain buried ice (they typically contain very low clay fractions, for instance). The (albeit restricted) literature on shallow groundwater flow in deglaciating environments which actually suggests very low degrees of surface hydrological connectivity and hence overland flow generation – Tom Müller's work is a good starting point (e.g. in HESS). So do these environments produce overland flow and soil erosion? What is the state of the soil development processes in this river basin that may lead to higher rates of moisture retention and overland flow generation? Where has soil developed? What is the link to buried ice (in the most upper parts of the catchment) which can be a crucial generator of impeded drainage and surface saturation? What evidence is there that processes matter besides basic fluvial reworking of former glacial sediments in high discharge events? There are many processes that could explain the results obtained and the review needs to be more balanced as there is a gap between the high quality analysis and the interpretation presented. Two things are required in revision; (a) a much more thoughtful and careful use of supporting literature, refocused on what we know of the geomorphology and hydrology of these kinds of Alpine landscapes, both in the introduction and the discussion; and (b) much more circumspect interpretation as there is only so much you can do to disaggregate catchment-scale yield data to identify potential sources within the river basin.

> *Reply: Thank you for this constructive critique. We agree that our interpretation must be grounded in glacial–paraglacial process understanding in addition to rainfall-driven erosion. In revision we will:*

1. *Refocus the literature: We will substantially reduce or remove citations that are not directly transferable (e.g., Kampf et al. (2016), Shen et al. (2016)) unless explicitly framed as conceptual background, and we will expand coverage of Alpine-relevant work (e.g., Swift, Gurnell, Hannah; studies on shallow subsurface flow and limited surface connectivity such as Müller; and recent paraglacial process studies).*

2. *Interpretation and discussion of processes: We will rework the discussion of processes to be more nuanced, and revisit the argumentation around overland flow and related erosion processes.*

   *While the referee makes an important point about infiltration rates and shallow groundwater in unconsolidated sediments, in the recently de-glaciated areas of the Vent-Rofental basin (where the authors incidentally have been for fieldwork on several occasions) the sediments can also have quite high fractions of clay and silt. Therefore we do not find it unreasonable to assume that runoff generation through infiltration-excess can play a role during heavy precipitation events. There is furthermore visible signs of rill and gully erosion within the catchment.*

   *Also, the authors have, in fact, spent much time in the study area, also during a heavy precipitation event (more specifically event 2023-g).*

3. *Adopt a more circumspect tone: We will emphasise that catchment-scale SSY integrates multiple sources and pathways, limiting source attribution. Where we cite studies from other environments, we will state why they are relevant (or note their limitations) for the Alpine setting.*

**4.** Point (3a) needs to be supported by better river basin characterisation. For each station we need to know contemporary % ice cover (we are just told it is 10% for whole system what is it for Vent). It would also be good to know what % of the catchments were ice covered during the Little Ice Age and how vegetation cover has developed since, given its potential impacts on erosion processes. This would help us to get a better understanding of process regimes associated with the basin.

*Reply: Thank you for this helpful suggestion. We agree that a clearer basin characterisation will strengthen the interpretation. Previous work in the basin by Schmidt et al. (2022) has examined this in more detail, and we will add the major points to enable the reader to better interpret the results presented.*

*We suggest to add a table with glacier cover percentages for both the whole Ötztal and the Vent subcatchment using available inventories (LIA, 1960, 1998, 2006, 2015). We will also quantify the area deglaciated since the LIA, and, if feasible, show this in Figure 1. In addition, we will add information on current vegetation and sparsely vegetated surfaces to better illustrate spatial contrasts in surface stability and erosion potential.*

*Schmidt, L. K., Francke, T., Rottler, E., Blume, T., Schöber, J., & Bronstert, A. (2022). Suspended sediment and discharge dynamics in a glaciated alpine environment: identifying crucial areas and time periods on several spatial and temporal scales in the Ötztal, Austria. Earth Surface Dynamics, 10(3), 653–669. https://doi.org/10.5194/esurf-10-653-2022*

**5.** The analytical approach in Table 2 is a bit weak. Your SSY data are effectively the sum of SSC*Q. Q is directly driven by precipitation. So, you would expect strong P versus SSY correlations. What is perhaps more interesting is to look at when P and SSY is not correlated – this

might say much more about what is going on. Note, I also struggled to understand which datasets were being used for Table 2 – Tumpen or Vent?

> *Reply: We agree that, since SSY largely reflects the product of SSC and discharge, and discharge itself is closely linked to precipitation volume, a correlation between P and SSY is to be expected. Nevertheless, we consider it important to show both yield and flux, in addition to peak concentration, as these variables reflect different aspects of catchment response. We will also add average SSC to Table 2 for completeness.*
>
> *What we find particularly interesting and what Table 2 is intended to highlight, is that it is important to consider rainfall and not just precipitation. This is not a trivial point in our setting, because part of the precipitation even in summer can occur as snow, which contributes little to immediate runoff and sediment mobilisation.*
>
> *Furthermore, we want to show that rainfall intensity (both peak and mean) is more strongly associated with the sediment response variables than total rainfall amount. The pattern that rainfall peak and average intensity correlate more strongly with both SSY and peak SSC suggests that it is not simply the total water volume that matters. This also holds for peak SSC, which does not explicitly include the discharge (although it is of course not wholly independent, given the positive relationship between discharge and SSC in the catchment).*
>
> *We will make these points clearer in the manuscript both in the results and discussion.*
>
> *We acknowledge that we omitted to specify the dataset source; Table 2 refers to Tumpen and we will clarify this in the revised manuscript.*

**6.** Figure 6. Some of these relationships look like a linear fit it being done to a non-linear dataset. This is really difficult to see with these plots, especially with the marker being scaled by even[t] duration. Non-linear relationships here might tell you a lot.

> *Reply: Note that in the linear regression, the sediment response variables, i.e. the dependent variable, has been log10 transformed. We have examined the residuals for each linear regression model, and they are normally distributed (see figure below). Therefore, find it reasonable to apply a linear regression model. The figure will be added to supplementary materials.*

---

## Author Comment (AC2)

**Anonymous Referee #3 from 24 Oct 2025**

https://doi.org/10.5194/egusphere-2025-3683-RC3

This study builds on years of meteorological, hydrological and turbidity/suspended sediment load (SSL) data collected by different institutions in and in the vicinity of a large partially glaciated catchment in the Austrian Central Alps for an event-based analysis of the contribution of different types of rainfall/precipitation/discharge events to suspended sediment yield. The study area Ötztal includes the smaller and more glaciated subcatchment Vent-Rofental. For the definition of precipitation events, the authors develop an interesting multi-scale approach, both spatially (catchment-scale vs. "grid" scale) and temporally (event duration). From the latter approach, they derive, for every duration class, a threshold at the respective 80% exceedance probability above which heavy precipitation events are delineated and characterized by a number of parameters. The authors then investigate trends in event number and characteristics, and the contribution of different types of events to the annual sediment yield. Major findings include an increase in the frequency of heavy precipitation and contrasting trends of annual suspended sediment yield that are attributed to different timing of 'peak sediment' and different reaction to heavy precipitation in the more glaciated part of the valley.

Regarding the study region and the focus on (changes in) suspended sediment load, the study connects with earlier publications by the same working group (Schmidt et al), but represents an original contribution. We appreciate the methodological development for the event-based analysis (and uncertainty assessment) of the INCA, discharge and SSC data that could be transferred to other study areas where comparative spatially distributed precipitation/rainfall data are available.

The manuscript is very well written and contains very rich figures, some of which might take readers some time to understand all the details contained in the diagrams. We think that the study is highly interesting for the geomorphological and hydrological scientific community and can be published pending moderate revisions being made.

**General Comments**

- (over)use of the term 'extreme'. We suggest to replace this by "heavy precipitation event" whereever applicable (see e.g. L169 – the 10 mm/h judiciously mentioned there as "heavy" are for sure not extreme; L193: catalogue of heavy precipitation events instead of "precipitation extremes"). We acknowledge that extreme value statistics play a role in the analysis and that the results are used as a detection threshold for such events. However, not every single one of the events listed in the final catalogue are truly extreme in light of the whole dataset and the definition of extreme as very rare (an event with a return period of 1,25 years is for sure not to be termed extreme) and of the M sample as annual maxima that don't really need to be 'extreme' in a sense different from just being the highest in one year.

  *Reply: We agree that using the term "heavy precipitation event" makes more sense and will make adjustments.*

- "grid scale" vs. "catchment scale": While 'catchment scale' can be understood intuitively, we suggest to rename "grid scale" (a grid can represent a whole catchment as well; it

consists of grid cells) in order to better represent the 'grid cell' or even more simply 'local' scale.

*Reply: We see the point about "grid scale" not being an intuitive term. However, "local" can carry many meanings, and could be interpreted differently. We will adopt the term "grid-cell scale" instead.*

- Units seem to be formatted differently than normal text. If that is intentional and typesetting rules requires that => OK

*Reply: Comes from the typesetting specific to the LaTex template.*

- Figure captions: We do appreciate the very rich figures, but in some cases we'd suggest to facilitate the readers' orientation and intuitive understanding by including more direct relationships between axis title/label colour and diagram content, legend etc (see specific comments) instead of having lots of additional information in the figure captions.

**Specific comments:**

L 29: While this might be the finding of the cited studies, the reader might stumble across that statement because they've just read in the abstract that one major finding of this study is a decline in annual sediment load at Vent-Rofental! We suggest to tune that down to "some studies have found a measurable increase…"

*Reply: The fact that the a measurable increase are not found everywhere is already clear in the phrasing: "measurably increased the amount of fluvial sediment exported **from some high-mountain areas**"*

L51f: The existence of high(er) amounts of unconsolidated sediments and sparse vegetation cover "downstream of glaciers" is only part of the paraglacial morphodynamics theory. According to the latter, it's not only the existence and amount of sediments, but also the topographic and lithologic characteristics of hillslopes exposed by deglaciation. So, first, "downstream of glaciers" does not define well the proglacial areas (that include specifically important hillslopes) that are affected by paraglacial dynamics – we suggest to replace "downstream of glaciers" with "proglacial areas exposed by deglaciation". Second, one characteristic of paraglacially enhanced erosion/reworking/transfer of sediments is that it is "system-internal", not requiring increased external forcing. Hence, an increase in heavy precipitation would be anticipated to further enhance and maybe accelerate paraglacial dynamics.

*Reply: We will adopt the suggested phrasing.*

L58f: We feel that SSL/SSC peaks and especially hydrological events that are not driven by (heavy) precipitation (which you do address in the study!), need to be mentioned here already.

*Reply: Will do.*

L68ff: Start with a sentence that makes it clear(er) that you have a nested catchment approach, i.e. that you investigate the whole catchment (with characteristics such as glaciated proportion) AND the Rofental subcatchment (with much more glaciated area). The two are easily defined as the contributing areas of the two gauging stations, so reference to the nested approach also introduces 2.1 well.

*Reply: We will make the nested approach clear from the first sentence of Section 2.*

L70-72: A bit confusing: (a) 10% of the (whole?) catchment are currently glaciated => need to give values for the whole vs. the sub-catchment. (b) "Glacier volume is projected to 4-20% by 2100": 4-20% refers to what initial volume? Present-day? "pre-industrial"?

*Reply: Reviewer 1 also asked about the glacier coverage and its evolution. We will include a table which shows the glacier coverage for both basins for each of the available glacier inventories. We will also include a mention of the change since the Little Ice Age, and if Figure 1 allows, highlight the areas deglaciated since the LIA.*

L80-86 refers more to the catchment (and the subcatchment); consider moving this to the introductory "study area" paragraph, while 2.1 would then focus on the measurements conducted at the gauging stations.

*Reply: We will intregrate the descriptions of the two gauging stations and catchments to the opening paragraph of section 2.*

Fig1: The Rofental catchment boundary is hardly visible in the big map (c). Better visibility would support the 'nested catchment' approach that we ask to make explicit in the text. Using a different colour than gray in (b) and (c) would probably do the job in both maps.

*Reply: We will change the colour of the outline.*

Fig1 caption: "The topography of Ötztal is steep" => consider adding a table that gives e.g. average steepness and other characteristics (such as percent area glaciated see previous comment)

*Reply: We will include this in the aforementioned table with the glacier coverage.*

Fig2: We suggest to make the Fig as large as possible (text width); it is quite dense (which is fine, other figures are even more so) and therefore maximum size is needed to help the reader.

*Reply: We will increase the figure size.*

L92: Use "suspended sediment concentration" at the first mention of SSC (followed by a comma and SSCt). Add information on the 'missing step' from the originally measured turbitidy to SSC. The single "t" in line 93 can be removed, the unit is just "15min-1").

*Reply: The first mention of "suspended sediment concentrations" is the paragraph before, where the acronym is defined. However, it might be clearer to not use the acronym here, so we will adopt the suggested change.*

*We will add more details on the turbidity-derived SSC in L87-88.*

*We suggest to change to "tonnes per time step" instead.*

L95 …are hourly precipitation grids" consider adding "at XX km resolution" (same: abstract, L9) and the number of weather stations to "from weather stations"

*Reply: We will add these clarifications.*

2.1: We appreciate the detailed explanation of INCA and the own validation approach of this paper. Consider moving the "quality check" (L112ff) to the methods chapter (3.1) uncertainty analysis

*Reply: We prefer to keep the mention of the "quality check" here as it related to the merging of the two INCA sets with different resolution, and not to the uncertainty analysis.*

L145ff: Does the computation of deviations between INCA and OBS include "0 precipitation" hours? Further down a threshold is introduced to distinguish wet from dry days, which is good - but that does not refer to the amount of rain in INCA vs Station data. We suppose ME and RMSE are likely biased by including a large number of dry hours where the deviation between INCA and station are 0. This is either a point to discuss in the discussion section or preferably to change (use only hours with nonzero precipitation in the station record)

*Reply: Yes, the computation of RMSE and ME shown in Fig. A3. We also checked the RMSE and ME for wet-days. The results are not substantially different as can be seen by comparing the figure below to Fig. 5.*

*We suggest to include this figure in the supplementary materials, and make mention of the key results in the results section.*

[Figure]

L177: Consider referring to Fig in Appendix

*Reply: Will do.*

L181f: Consider adding one more sentence so that the reader does not have to look up Ulrich et al. The GEV distribution family has three parameters (location, scale, shape), - how many (which) parameters are being fit in the dGEV and why can the number of parameters be reduced?

*Reply: The number of parameters are reduced, since if a duration-dependent GEV is not used, you have to fit a GEV to each duration separately each with 3 parameters, which for 8 different durations are 8 * 3 = 24 parameters, whereas using a duration-dependent GEV with an additional multiscaling parameter as we used can fit a single 6-parameter distribution for all duration simultaneously.*

*We will add more clarification on the duration dependent GEV to the text.*

L190: Here, you name the "0.8 exceedance probability quantile", while in the caption of Fig A2, you write "0.2 non-exceedance probability". Pls homogenise.

*Reply: We will update to use the term "0.2 non-exceedance probability" throughout.*

L187ff: This makes one think immediately of runoff/discharge events triggered by snow (and glacier) melt without precipitation. Yes, snowfall in winter is not relevant to your study, but snowmelt in spring is.

*Reply: True, however, this portion of the text concerns itself with precipitation and it's phase. Reviewer 2 requested more consideration and discussion of melt-driven processes and this will be incorporated more into the manuscript.*

L194f: The whole "peak detection" approach is explained in the caption of Fig. 3: While that enhances the immediate understanding of Fig3, it's sort of missing in the text. If you like to keep it as is, consider writing "…tpeak (Fig. 3, explained in detail in figure caption)"

*Reply: We will add the reference to the caption.*

Eq5/6, L198ff: We appreciate your approach (starting from peak, searching for first and last threshold exceedance before and after the peak, respectively) as an alternative to the "peak over threshold" declustering approach (that starts from a threshold exceedance) used in the corresponding extreme value statistics. However, what happens if the threshold is not exceeded for just one 15 min data point before exceeding it again? In some declustering approaches, a user-specified parameter would prevent two events from being separated by just one (especially that short) 'break' in a time period otherwise exceeding the threshold. We feel this could be explored and, also if not implemented, at least discussed later.

*Reply: When detecting the extreme peaks over a threshold, we used the pyextremes.get_extremes function which includes declustering of peak over threshold (POT) extremes. We used a minimum time distance (window duration) between adjacent clusters of 24 hours. Note that the minimum time distance parameter will only have a limited influence, since we then (1) subsequently searched forward and backward in time to isolate the whole precipitation event, and (2) merged overlapping events. We tested minimum time distance parameters, e.g. the same as the threshold duration, 12 hours, 6 hours, etc. The number of events detected varied minimally (one or two events more or less) or not at all. This suggests that the isolation of the whole precipitation event and merging overlapping events has much greater effect than the minimum time distance parameter for the de-clustering of peaks.*

L215: Consider giving an example of what kind of pattern would be judged as a data artefact/mistaken detection

*Reply: We will add an example.*

Headline 3.3: Would "Classification and characterisation of heavy precipitation events" be better? We felt yes because you do not only characterise the events but also categorise/classify them

*Reply: We will update the heading.*

L235: Not clear to what the "search window" refers, pls specify. It is not possible to imagine what constitutes a hydrological event (as opposed to the precipitation event whose delineation is described in detail). This has implications for the following, specifically for the hydrological events that are not triggered by (heavy) precipitation.

> *Reply: The "search window" refers to a parameter for the local minima algorithm. A 21 hour window means each local maximum is at least 10.5 hours apart. A hydrological event is the slice of the streamflow timeseries between two local minima. As such a hydrological event is a streamflow pulse. We will update the text to make this clearer.*

Eq 7,8,9: add units for SSF and SSC

> *Reply: Will do.*

L255f: The section heading specifies "precipitation-driven events", but how does the approach detailed here deal with hydrological events that are not triggered by precipitation but by snow and/or glacier melt? Fig 2 clearly shows that spring snow-melt is relevant specifically at Tumpen. Pls address this in more depth in the methods and/or discussion section.

> *Reply: As mentioned above we will include more discussion of melt-driven processes.*

L260: typo: Theil-Sen slope

6, L 262ff: This is about "sediment discharge events". How are these detected? Do they always coincide with hydrological events? That is, are "sediment discharge events with high SSC / SSC spikes" exclusively a subset of hydrological events? We can't know, but couldn't it be that, for some reason, a sediment spike takes place in a minor hydrological event that does not belong to the sample/catalogue of hydrological events? E.g. a very localised rainfall event that does not lead to highly increased discharge but to a substantial input of suspended sediment? Or the sudden mobilisation of subglacial sediment without a very conspicuous hydrological event recorded at a gauge kilometres away? Is it possible that a "peak SSC event" is longer than the hydrological event(s) it is attributed to?

> *Reply: Yes, it is assumed that a sediment discharge event is a spike in SSC contained within a hydrological event, or explained differently: a hydrological event where the peak SSC is above the 90th SSC percentile.*
>
> *Since the hydrological event catalogue covers the entire period with streamflow and SSC data (given that it splits the streamflow time series at local minima), there is no part of the study period that is not covered by a hydrological event. Therefore, we will never miss a SSC spike outside of a hydrological event, because every SSC spike will be contained within a hydrological event.*
>
> *There is a possibility of the streamflow pulse (i.e. hydrological event) not being closely linked with the suspended sediment pulse recorded at the same time. However, given the strong influence of streamflow on SSC in Ötztal, and especially the pronounced diurnal cycle in streamflow, it is unlikely that this type of event actually occurs within the catchments.*

1, L269ff: See comment @ methods: Precipitation=0 included in uncertainty assessment?

> *Reply: See reply above.*

L277: INCA under-predicts => suggest to use "underestimates" instead of "predicts"

> *Reply: Will do.*

L285: This is an example of "extreme" that we'd like you to reconsider

Fig 4 caption/legend:

- Dot size indicates the duration of events, but there's no legend item. Or is dot size meant to be only a qualitative / ordinal measure of duration?

  *Reply: Yes, it is meant to be qualitative.*

- in addition to the event with the highes Ptot and Imax. Event 2020-k...: These three events are not shown in the figure – which is a bit confusing. Information on single events not shown should be in the text.

  *Reply: All events mentioned are shown in panel b which is what this part of the caption in explaining.*

- Average precipitation area: Is a relative measure (related to the total catchment size)? Shouldn't it be called "relative precipitation area" instead? The same applies to section 3.3 L220f.

  *Reply: Yes, it is a relative measure. But calling it the "average relative precipitation area" is confusing. Furthermore, the measure is explained in section 3.3.*

- Especially in (c), the Vent/ROfental catchment boundary is barely visible (see earlier comment), consider choosing a different colour.

Fig5a

- FB is hard to see (dark purple on dark grey), consider choosing a different colour

- Add a pink left y axis label for RMSE

- Consider making the precipitation y axis labels blue in order to ease relation to the blue line (like RMSE and also for number of events).

- We suggest to specifiy "Precipitation" axis title instead of just explaining in caption => "May-Oct precipitation [mm]"

  *Reply: We will adopt the suggested changes.*

Fig5b

- Dot size should have a legend item

- Include grey dots/circles in the legend (otherwise, in order to understand the figure, the reader would have to read the whole caption first)

  *Reply: We will expand the legend to make it clear what the dot size and colour refer to, although, we do not think it unreasonable for the reader to read the caption in order to understand details from the figure. That is, after all, what the caption is for.*

Tab 2: add number of events (n) and type of events (all?). Moreover: Does r refer to Pearson's r? If so: Pearson's r only quantifies linear correlations, Spearman's r would quantify all kinds of (monotonous) correlations.

*Reply: We will consider to use the Spearman's correlation coefficient and update the legend to make it more clear that the numbers refer to the correlation across all events and not by class.*

L300ff: We accept that your MK test was significant, and that RMSE does not greatly vary between the two major parts of your data characterised by different (INCA vs. pre-INCA) data available. However, can you exclude that the MK trend rather represents a difference between the two parts that is (partially) not due to climate change but to a change in how the data were acquired?

*Reply: No, we cannot exclude this possibility. This is already part of the discussion. At the prompting of another reviewer, we have also looked at the event numbers at stations within the catchment, and found that there is an increasing trend at all of them, although not significant for all. This will also be added to the discussion.*

L324: The difference is particularly pronounced for RFmax and SSY: We think RFmax has been confused with RFtot – here, the difference in r between SSY and SSF is 0.51 vs. 0.27, while it is much more similar for RFMax and SSY/SSF…

*Reply: The difference referred to here is the difference between sub-daily and long-duration extremes and it refers to Figure 6, not Table 2. This is quite clear from the reference to Figure 6 in parenthesis at the end of the sentence.*

Fig6: Consider adding a legend for point/circle sizes, and add "n" to the boxplots (has been done with other boxplots, should be the same here). Fig6 caption: "Labelled events" Only three out of five labelled events are mentioned in the text (2019g, 2020k, 2022d), and events are mentioned in the text that are not contained neither in Fig4 nor here (2020j,2020n, see page 24).

*Reply: We will add the number of events in each class to the top of the box plots. The labelled events are meant to be consistent with the ones shown in Fig. 4. We will consider adding more labelled events, but we also want to be cautious of cluttering an already detailed figure with too much.*

Fig7: Try to separate column a/c better from column b/d

L348: Remove duplicate "only about 10%"

*Reply: Will do. Thanks for spotting it.*

Fig8: We suggest to position the two plots horizontally (would make the figure fit better with page)

*Reply: This will be a single column figure, which will appear different in the final typesetting.*

L375: Pls specify what "geomorphological variables are sensitive to small-scale changes in rainfall spatial structure" means

*Reply: We will add some clarification.*

Lines 376ff somewhat repeat lines 370ff

L417ff: We suggest to include the findings of a study that conducted sprinkling experiments on steep moraines; these could be more representative of (parts of) your study area: Maier, F., Lustenberger, F., & van Meerveld, I. (2023). Assessment of plot-scale sediment transport on young

moraines in the Swiss Alps using a fluorescent sand tracer. Hydrology and Earth System Sciences, 27(24), 4609–4635. https://doi.org/10.5194/hess-27-4609-2023

> *Reply: Thank you for the suggestion.*

L432: add where the debris flows reported by the two papers took place (Horlach valley, tributary to Oetztal)

> *Reply: Will do.*

L468: See comment regarding the MK test results (L300) in light of the higher RMSE in the first years. In light of this, the "robust" in L525 of the conclusion needs to be reconsidered.

> *Reply: We will remove the word "robust" from the conclusions.*

Fig A2: Title of (b) should read "Local-scale (a placeholder for an alternative to "grid scale") maximum precipitation It, just like in the caption. Moreover, the blue threshold line (and others) use the "20% non-exceedance probability" unlike the "80% exceedance probability" terminology used elsewhere. Pls homogenise.

> *Reply: Already addressed in replies above.*

Fig A3 – Legend for "Station in Ötztal": Change fill colour to white, because the Ötztal stations have different colours and a thick black outline. Moreover, grey dots have been used elsewhere. Moreover, consider adding a legend for circle sizes

> *Reply: We think on the whole the figure legend is clear. However, we will add some more details to the caption to avoid misunderstandings.*

Fig A4: Legend for point/circle sizes missing; add n to the boxplots. Similiarly to Fig6, you could add number of events and time period.

> *Reply: We will adopt the figure to conform with suggested changes made to Figure 6.*

---

## Author Comment (AC3)

**Anonymous Referee #2 from 22 Oct 2025**

https://doi.org/10.5194/egusphere-2025-3683-RC2

**General comments**

This manuscript addresses the critical and timely topic of suspended sediment dynamics in a rapidly deglaciating Alpine catchment under the influence of increasing extreme rainfall events. This work is highly relevant to the scope of Hydrology and Earth System Sciences (HESS) as it provides quantitative insights into how climatic shifts are altering hydro-geomorphic processes in high-mountain environments. The study is well-conducted and presents a thorough analysis linking extreme precipitation characteristics to suspended sediment fluxes in the Tumpen-Ötztal and Vent-Rofental catchments in Austria. The methods are robust and clearly described, and the resulting data analysis is systematic. The manuscript is well-written, and the discussion is generally comprehensive. Overall, I recommend publication of the manuscript following minor revisions. Some comments are listed below.

**Specific Comments**

*Antecedent Catchment Conditions.* The discussion (lines 484–486) highlights the role of sediment availability for interpreting the year-to-year variability in SSY. In my opinion, authors should also consider the influence of antecedent catchment conditions, specifically factors like soil moisture, which might impact both overland flow generation and soil erosivity, the sequence of events, which dictates the depletion of readily available in-channel and hillslope sediment supply, and the presence/absence of snow cover, which might modulate the rainfall-SSY relationship. An event-based analysis of few targeted events could provide insights on these factors.

> *Reply: As the reviewer suggest, we will include information on (1) antecedent precipitation to give an indication of soil moisture, and (2) snow conditions in the catchment during events. Snow cover could well play a role in spring early and early summer, and we will include some discussion on this. Given the size of the catchment, and that many events are small scale only affecting some parts of the catchment assessing the depletion of sediment stores will be challenging. We will, however, have a look for cases where large scale events occur in sequence.*

*Snow and Ice Melt Processes.* In general, little space is given in the discussion on the impact of snow and ice melt processes on sediment transport. I suggest the authors expand this discussion to address the temporal dynamics and potential overlap with rainfall extremes. Specifically: When does the peak ice melt happen? Is it overlapping with the period characterized by the highest short-duration convective rainfall? Could this interaction explain the higher SSFmean observed for short-duration events? Furthermore, describing a typical pattern of snow cover duration in the catchment would add context. Again, by analyzing a few events with different characteristics, as they did for 2020, the authors would be able to incorporate these key cryospheric processes more fully into the discussion.

> *Reply: We will include information about the seasonality of glacier melt, snowmelt and snow cover in the description of the study area.*

*As for whether the short-duration events have higher SSFmean because they occur in the months with high ice melt, we propose to include the excess SSY of each event. This substracts the "base" sediment load at the beginning of the event before calculating the SSY, and then SSF excess can be calculated.*

*Event classification.* I have a clarifying question regarding the event classification described in Lines 224–226. Does this methodology imply that a genuine long-duration event could be identified or partially characterized as a sub-daily extreme if it contains a single, very intense sub-daily peak? I am not sure I understood this and I wonder how potential mis-classification might influence the results showed in Figure 6 as well as the discussion in 5.2.2. and 5.2.3.

> **Reply:** *Sub-daily extremes exclusively have detected extreme peaks above the 1 to 12 h threshold. This means that if an event has event a single extreme peak above a 24-, 48- or 72-hour threshold, it is classified as a long-duration event. So there is no case where a long-duration event might be classified as a sub-daily extreme.*
>
> *However, there might be let us say an event was extreme at the 6-, 12- and 24-hour duration. That event under our classification that event would be labelled as a long-duration extreme. We would not say this is a misclassification of said event, since it was extreme at a longer duration in addition to the short ones.*
>
> *We felt that keeping the purely sub-daily extremes together in one category was the cleanest way to separate the events into two categories. However, as we introduce in section 5.2.3. some events classified as long-duration ones, also have extreme peaks detected above the 1 to 12 hour thesholds.*
>
> *Using this three way classification, with "purely" long-duration, mixed long-duration and sub-daily extremes, may have implications for the linear relationships showed in Figure 6.*

*Increasing frequency of extreme precipitation – Stations.* The finding of an increasing trend in the frequency of extremes derived from the INCA product is central to the study's context. Have the authors checked if a similar increasing trend is observable in the precipitation station data used for INCA development? Recognizing the already thorough nature of the analysis, I suggest the authors check if a similar increasing trend is observable in some targeted stations. This would enhance the robustness of the signal by ruling out the possibility of the trend being an artifact of the gridded product or its calibration process.

> **Reply:** *This is a good suggestion. We have checked the trend in the number of events detected (with the same methodology as for the catchment time series). We selected 3 Geosphere Austria stations that are used in INCA (see Haiden et al., 2011). Two of these stations, Obergurgl and Umhausen are situated within the Tumpen-Ötztal catchment. The Pitztaler-Gletscher station is situated in a neighbouring catchment, but was included as a high elevation station, that is situated quite close to the watershed boundary.*
>
> *It is important to note that precipitation data from stations are only representative of the area immediately around the station. Thus the number of events are not directly comparable to the events detected from the catchment time series, as the station data cannot take into account the spatial scale of events, and might also miss extreme events occurring within the catchment.*

*Below we have compared the number of events detected at the stations each year including the trend over the same 20-year period as for the Tumpen catchment (2004-2023). The trend in the number of events (Mann-Kendall test with 5% significance level) is also shown. Umhausen station shows a significant increase in the number of events. The other two stations also show an increasing trend, but which is not significant.*

[Figure]

**Technical Corrections**

Figure 2. The font size is rather small. The figure could be a bit bigger.

> *Reply: We will increase the figure size.*

Line 100-101, 108-109: In the sentence "The accuracy of INCA estimates can vary, particularly in complex terrain, with an average error of 50-100% in the 15-minute precipitation grids and 1.0 to 1.5 ∘C in the temperature grids (Haiden et al., 2011).", the 15-minutes precipitation grids confuses because in line 100-101 the authors describe the INCA datasets as "… hourly 1-km grids for all of Austria.". Perhaps just add "… and sub-hourly …" to the sentence in line 100-101.

> *Reply: Haiden et al. describe the INCA data with 15-min resolution in their analysis, whereas we used the hourly resolution, which the freely available resolution. Since the resolution of the data matters for the error metric we felt that it was important to make the difference in resolution clear here. We will add a clarification to the lines 100-101.*

Line 115-116: Please, provide a brief (few words) explanation of the rainfall/snowfall separation method used in openAMUNDSEN.

> *Reply: We will add a brief explanation.*

Line 120-123: As I understand here, you used hourly, 1-km grids, precipitation for the period 2004-2024, and rainfall for the period 2011-2024. Correct? Please, clarify.

*Reply: Correct. As clearly stated in lines 117-119:*

*"As rainfall estimates rely on temperature grids (GeoSphere Austria, 2024a), which begin on 15 March 2011, **hourly rainfall grids are only available for the same time period as temperature (i.e. March 2011 to December 2024)**."*

Line 125. Figure 1 to Figure 1b.

*Reply: Will do.*

Line 190: Please improve clarity by changing "Detection thresholds, u, for each …" to "Detection thresholds, u, for each DURATION d and spatial scale (i.e. GRID-SCALE It or CATCHMENT-AVERAGED Pt).

*Reply: Will do.*

Line 210: I believe I understood what you did, but could you please write this iterative merging in a clearer way?

*Reply: The merging was done in several passes. Will try to write it more clearly.*

Line 265: Since you are talking about events with SSC larger than the 90th percentile, I find P90(SSCt) a confusing definition and would change it to SSC90.

*Reply: Will do.*

Figure 8. Font size rather too large. I suggest being more precise in the legend: from "extreme" to "extreme precipitation", from "non-extreme" to "non-extreme precipitation".

*Reply: Will adjust the font size. As for the legend labels, we prefer to keep these consistent with Figure 7 (that needed these abreviated legend labels to not get cluttered). The classification is also described in the caption.*